# CompSGD: Robust Comparison-Based Approach for Zeroth-Order Optimization under $(L_0, L_1)$-Smoothness and Heavy-Tailed Noise

## Abstract

In modern non-convex optimization, more and more attention is drawn to zeroth-order problems where the only available information is which set of model parameters is better, without quantitative characteristics. The data in these problems can be extremely noisy, and the models themselves are so complex that the standard smoothness assumption fails to describe them. Motivated by these challenges, we propose new zeroth-order methods to deal with generalized $(L_0, L_1)$-smoothness and severe heavy-tailed noise with bounded $\kappa$-th moment. Using only comparisons of function values at two different points, our MajorityVote-CompSGD method achieves the first-known high probability bound $\tilde{O}\left(\frac{\Delta \sigma^2 d^{9/2}}{\kappa^2}\left(\frac{L_0^3}{\varepsilon^6} + \frac{L_1^3}{\varepsilon^3}\right)\right), \kappa \in (0, 2]$ for number of comparisons under symmetric independent noise. If function values are available, our minibatch-CompSGD can converge to the desired average gradient norm after $\tilde{O}\left(\Delta \sigma^{\frac{\kappa}{\kappa-1}}(\frac{d^{3/2}L_0}{\varepsilon^2} + \frac{d^{3/2}L_1}{\varepsilon})^{\frac{2\kappa-1}{\kappa-1}}\right), \kappa \in (0, 2]$ function evaluations. In addition, we provide convergence guarantees for Lipschitz noise, parameter-free tunings and in expectation bounds with milder $d$ dependence.

## 1 Introduction

In many practical optimization tasks, the computation of the function gradients is infeasible, for instance, due to enormous sizes, non-differentiable function structure or lack of information. For such scenarios, methods that operate only with function values or even comparisons of these values are an active area of research (Nozawa et al., 2025; Jiang et al., 2024; Liu et al., 2024; Chen et al., 2024; Yin et al., 2024; Tang et al., 2023; Ouyang et al., 2022). Although modern applications, especially in deep learning, are known to exhibit heavy-tailed noise and very complex model structures, most related works consider only simple model descriptions and bounded variance noise, or no noise at all. This leaves an important practical problem without a proper theoretical framework. The goal of this work is to bridge this gap by introducing a new robust and theoretically justified method that uses only function values or their comparisons for optimization. We formally present the problem statement (§1.1), review related works (§1.2), and detail our contributions (§1.3).

### 1.1 Problem statement

Consider the following non-convex stochastic optimization problem:

$$\min_{x \in \mathbb{R}^d} f(x) := \mathbb{E}_\xi[f(x, \xi)], \tag{1}$$

where the random variable $\xi$ can only be sampled from an unknown distribution. The main goal is to find a point with the smallest gradient norm. For example, in machine learning, $f(x, \xi)$ can be interpreted as a loss function on a sample $\xi$ (Shalev-Shwartz & Ben-David, 2014).

We consider two oracle types: the **zeroth-order oracle** which, for any two points $x, y \in \mathbb{R}^d$, gives their noisy function values $f(x, \xi_x)$ and $f(y, \xi_y)$, and the **comparison oracle** $\phi(x, y, \xi := \{\xi_x, \xi_y\})$ which determines which noisy function value is larger:

$$\phi(x, y, \xi) = \text{sign}(f(x, \xi_x) - f(y, \xi_y)). \tag{2}$$

The realizations $\xi_x$ and $\xi_y$ can be both independent or depend on the points $x, y$. The comparison oracle is natural for describing human decision-making (Lobanov et al., 2024). Given a choice between two options, it is much easier to choose which option is better rather than estimate quantitive difference. The stochasticity $\xi$ describes a variety of decision makers and their random states. This oracle is extensively used in Reinforcement Learning (RL) and Large Language Models (LLMs) training via RL with human feedback (Ouyang et al., 2022; Wang et al., 2023; Tang et al., 2023).

$(L_0, L_1)$**-smoothness.** In early theoretical works, deep learning models were described using a standard smoothness assumption, namely, $\|\nabla f(x) - \nabla f(y)\|_2 \leq L_0\|x - y\|_2, \forall x, y \in \mathbb{R}^d$. However, a new generalized $(L_0, L_1)$-smoothness condition was recently introduced in (Zhang et al., 2020b) to characterize LLMs whose Hessian norm exhibits linear growth: $\|\nabla^2 f(x)\|_2 \leq L_0 + L_1\|\nabla f(x)\|_2$. The future works have extended this framework to only once differentiable functions and various settings including cases with symmetric and asymmetric power norm growth (Chen et al., 2023) and sub-quadratic polynomial growth (Li et al., 2023). Applications of the generalized smoothness can be found not only in training of LLMs (Zhang et al., 2020a; Liu et al., 2023b), but in distributionally robust optimization (Levy et al., 2020; Jin et al., 2021), multitask learning (Zhang et al., 2024), federated learning (Liu et al., 2022), bilevel optimization (Hao et al.; Gong et al.) and other domains.

**High probability bounds.** The remarkable success of stochastic first-order methods for optimizing neural networks (Bottou, 2012; Kingma & Ba, 2014) has inspired extensive research into the theoretical convergence properties of these methods under various noise assumptions. Early studies (Nemirovski et al., 2009; Ghadimi & Lan, 2013; Bernstein et al., 2018a) derive complexity bounds in expectation, typically under light-tailed noise such as sub-Gaussian or bounded variance (BV) noise. However, given the high computational cost of training large deep learning models (Davis et al., 2021), there is a growing interest in *high-probability (HP)* convergence guarantees (Sadiev et al., 2023; Nguyen et al., 2023; Hübler et al., 2024). Unlike bounds in expectation, which describe average performance across multiple runs, HP bounds ensure convergence for individual runs with probability at least $1 - \delta, \delta \in (0, 1)$. Although Markov's inequality can convert bounds in expectation into HP bounds with extra $\frac{1}{\delta}$ factor, the researchers strive to obtain bounds with tighter $\log \frac{1}{\delta}$ dependencies.

**Heavy-tailed noise.** Furthermore, recent studies indicate that mentioned BV assumption fails to capture the noise characteristics in modern deep learning tasks. For instance, in Transformer models, stochastic gradients often follow a *heavy-tailed (HT)* distribution (Simsekli et al., 2019; Zhang et al., 2020b; Gurbuzbalaban et al., 2021). It means that the noise in function estimates has a bounded $\kappa$-th moment for some $\kappa \in (1, 2]$. In the zeroth-order optimization, two main types of corrupting noises are considered: *independent* which corrupts each point $x$ individually and *Lipschitz* which corrupts a pair of points $x, y$ together and decreases as these points become closer.

## 1.2 RELATED WORKS

Zeroth-order optimization has a rich history including both discrete and continuous classic approaches such as Bayesian optimization (Shahriari et al., 2015; Balandat et al., 2020), Evolutionary algorithms (Lei et al., 2025; Salimans et al., 2017), ellipsoid methods (Bland et al., 1981) etc. The most relevant methods for optimizing non-convex smooth functions are based on finite-difference approximations.

**Zeroth-order finite-difference methods.** These methods utilize explicit function values to approximate gradient by a finite difference which is then plugged into the first-order methods (Nesterov, 2011; Ghadimi & Lan, 2013; Duchi et al., 2015; Shamir, 2017; Gasnikov et al., 2022). The gradient estimate $g_\tau(x, \mathbf{e})$ with the arbitrary smoothing parameter $\tau$ is built on a random direction $\mathbf{e}$ sampled from the unit Euclidean sphere:

$$g_\tau(x, \mathbf{e}) := d(f(x + \tau\mathbf{e}, \xi_+) - f(x - \tau\mathbf{e}, \xi_-)) \cdot \mathbf{e}/(2\tau). \tag{3}$$

For non-convex functions under Lipschitz BV noise, it is enough to use SGD with gradient estimate (3) (ZO-SGD) and sample $\mathbf{e}$ to obtain the rates $O(d\varepsilon^{-4})$ in expectation (Ghadimi & Lan, 2013). To cope with HT noise, the authors of (Kornilov et al., 2023; 2024) use more robust SGD with clipping of heavy-tailed gradient estimates (ZO-ClipSGD) and obtain HP bounds for convex functions. For independent noise, there exists a series of works dedicated to smoothed-based methods with Decision-Dependent Distributions, achieving rates $O(d^2\varepsilon^{-6})$ (Liu et al., 2023a) (for bounded optimized functions) and $O(d^2\varepsilon^{-4})$ (Hikima et al., 2025). Under $(L_0, L_1)$-smoothness, only paper (Lobanov &

Gasnikov, 2025) provides the linear bounds in expectation for the zeroth-order convex setup. The authors also observe considerable convergence boost for functions with $L_0 \approx 0$.

There exist even more challenging optimization problems in which only comparisons of function values are available. For these problems, the methods with a comparison oracle (2) come in handy.

**Comparison-based methods.** These methods use only comparisons without direct function evaluations. They find the minimal point among the observed ones following random directions. The most common instance is Stochastic Three Points (STP) method (Bergou et al., 2020) and its variation with momentum (SMTP) (Gorbunov et al., 2020). For the current iteration $x^k$, it takes a random direction $\mathbf{e}^k$ with stepsize $\gamma_k$ and goes along it where the function value is smaller (or stays):

$$x^{k+1} = \arg\min\{f(x^k), f(x^k + \gamma_k \mathbf{e}^k), f(x^k - \gamma_k \mathbf{e}^k)\}.$$

In these works, the methods are analysed for non-convex functions without any noise. In (Boucherouite et al., 2024), the authors work with sum-type functions and stochastic mini-batches. They prove sample complexity $O(d^3 \varepsilon^{-6})$ in expectation for STP under independent BV noise in function estimates, but with huge batch sizes $O(d^2 \varepsilon^{-4})$. In (Saha et al., 2021), the authors consider a independent noisy comparison oracle where noise is introduced as a fixed probability of receiving a wrong sign during the comparison. They restate STP via the sign operator and at each iteration repeated Bernoulli trials with comparisons to ensure the sign correctness with high confidence. The authors obtain HP bounds, but only for convex and strongly-convex functions.

We highlight the ZO-SignSGD method (Bernstein et al., 2018a;b; Liu et al., 2019a) which belongs to both method groups simultaneously. It takes only the signs of gradient estimates (3). For the current iteration $x^k$ and direction $\mathbf{e}^k$, its update step is:

$$x^{k+1} = x^k - \gamma_k \cdot \text{sign}((f(x^k + \tau \mathbf{e}^k, \xi_+) - f(x^k - \tau \mathbf{e}^k, \xi_-)) \cdot \mathbf{e}^k). \tag{4}$$

The sign operator from this update step can be computed by comparison oracle without direct function values. For non-convex sum-type functions with Lipschitz bounded noise, the authors of (Liu et al., 2019a) prove the zeroth-order sample complexity $O(d^2 \varepsilon^{-4})$.

In (Lobanov et al., 2024), the authors propose OrderRCD method which is combination of Coordinate Gradient Descent and the search for the steepest stepsizes using the golden ration method, which requires only comparisons. In (Tang et al., 2023), a comparison oracle is used to build a ranking-based gradient estimate over random directions, which then is plugged into GD.

***All the previous non-convex results for zeroth-order and comparison-based methods are proved under standard $L_0$-smoothness and mostly under BV noise.*** Meanwhile, real-world applications motivate to use more general assumptions on smoothness and noise, as we do in this paper.

### 1.3 CONTRIBUTIONS

**Theory.** We derive **the first-known high probability convergence bounds for non-convex zeroth-order optimization under generalized $(L_0, L_1)$-smoothness and HT independent or Lipschitz noise.** For standard smoothness, these results are new as well. See Section 2.

We propose our robust MajorityVote-CompSGD (Algorithm 2) that uses only function comparisons for optimization. To achieve accuracy $\varepsilon$ in average $\ell_2$-gradient norm, it needs the following number of comparisons under independent symmetric noise ($\Delta = f(x^1) - f^*$, $d$ — dimensionality and $\kappa, \sigma$ — bounded moment power and value, Theorem 1):

$$\tilde{O}\left(\left(\frac{\Delta L_1 d^{\frac{3}{2}}}{\varepsilon} + \frac{\Delta L_0 d^{\frac{3}{2}}}{\varepsilon^2}\right)\left[\frac{1}{\kappa^2} + \frac{\sigma^2 d^3}{\varepsilon^2}\left(L_1 + \frac{L_0}{\varepsilon}\right)^2\right]\right), \kappa \in (0, 2].$$

For zeroth-order oracle where function values are available, we present our minibatch-CompSGD (Algorithm 3) that under any independent noise has complexity (Theorem 2):

$$\tilde{O}\left(\left(\frac{\Delta L_1 d^{\frac{3}{2}}}{\varepsilon} + \frac{\Delta L_0 d^{\frac{3}{2}}}{\varepsilon^2}\right)\left[1 + \left(\frac{\sigma d^{\frac{3}{2}}}{\varepsilon}\left(L_1 + \frac{L_0}{\varepsilon}\right)\right)^{\frac{\kappa}{\kappa-1}}\right]\right), \kappa \in (1, 2].$$

Moreover, we provide convergence guarantees for Lipschitz noise, parameter-free algorithms tunings and in expectation bounds with milder $d$ dependence in the corresponding theorems.

**Experiments.**    To validate our theoretical findings in real-world scenarios with heavy-tailed noise, we evaluate sign-based methods on the fine-tuning of Transformer models, demonstrating their effectiveness in both NLP classification and image generation tasks. See Section 3.

## 2    HIGH PROBABILITY BOUNDS FOR COMPARISON AND ZEROTH-ORDER ORACLES UNDER HEAVY-TAILED NOISE AND GENERALIZED SMOOTHNESS

We begin this section by introducing the necessary assumptions (§2.1) and the backbone method CompSGD (Alg. 1, §2.2). Then we propose our MajorityVote-CompSGD (Alg. 2, §2.3) method for comparison oracle (2) to optimize $(L_0, L_1)$-smooth non-convex functions corrupted by symmetric and unimodal HT noise. We prove its HP convergence guarantees for the best parameters and also for *parameter-agnostic tuning*. In addition, for classic zeroth-order optimization with available function values, we propose minibatch-CompSGD (Alg. 3, §2.5) under HT noise without symmetry assumption. We discuss and compare our methods with related works in §2.4 and §2.6, respectively.

### 2.1    ASSUMPTIONS

We use the following assumptions on the objective function $f(\cdot)$ and noisy function estimates $f(\cdot, \xi)$.

**Assumption 1** (Lower bound). *The objective function $f$ is lower bounded by $f^* > -\infty$.*

**Assumption 2** ($(L_0, L_1)$-smoothness). *The objective function $f$ is differentiable and (symmetrically) $(L_0, L_1)$-smooth, i.e., for the non-negative constants $(L_0, L_1)$ and $x, y \in \mathbb{R}^d$, it holds*

$$\|\nabla f(x) - \nabla f(y)\| \leq (L_0 + L_1 \sup_{u \in [x,y]} \|\nabla f(u)\|)\|x - y\|. \tag{5}$$

For examples of $(L_0, L_1)$-smooth functions and its basic properties, we refer reader to Appendix A.1.

**Assumption 3** (Heavy-tailed noise in function estimates). *For two points $x, y \in \mathbb{R}^d$, the stochastic difference $f(x, \xi_x) - f(y, \xi_y)$ is unbiased estimate of the true difference $f(x) - f(y)$:*

$$\mathbb{E}_{\xi_x, \xi_y}[f(x, \xi_x) - f(y, \xi_y)] = f(x) - f(y),$$

*and satisfies one of the conditions below for $\sigma > 0$ and $\kappa \in (1, 2]$:*

1. **independent noise:** $\mathbb{E}_{\xi_x, \xi_y}[|f(x, \xi_x) - f(y, \xi_y) - (f(x) - f(y))|^\kappa] \leq \sigma^\kappa, \quad \forall x \in \mathbb{R}^d,$

2. **Lipschitz noise:** $\mathbb{E}_{\xi_x, \xi_y}[|f(x, \xi_x) - f(y, \xi_y) - (f(x) - f(y))|^\kappa] \leq \sigma^\kappa \|x - y\|_2^\kappa, \quad \forall x \in \mathbb{R}^d.$

The example of independent HT noise is the estimate $f(x, \xi)$ corrupted at each point by independent heavy-tailed noise $\xi$ with bounded $\kappa$-th moment: $f(x, \xi) := f(x) + \xi$. As instance of Lipschitz HT noise, one can use estimate $f(x, \xi) := f(x) + \langle x, \xi \rangle$ where $\xi$ - is $d$-dimensional HT noise. For a sum-type function $f(x) = \frac{1}{K} \sum_{i=1}^{K} f_i(x)$ with $\xi$ denoting a random batch $I$ from $\{1, \ldots, K\}$, the estimate is $f(x, \xi) = \frac{1}{|I|} \sum_{i \in I} f_i(x)$. In this case, the oracle gives the same $\xi$ realization (batch) for two points. This estimate can satisfy both independent and Lipschitz noise assumptions depending on function properties (Boucherouite et al., 2024; Liu et al., 2019a).

**Random directions.**    Usually zeroth-order methods first explore function changes along some random directions sampled from the chosen set $\mathcal{D}$ and then make next step. This set should be wide enough to capture the full information about function changes, thus, we assume the following.

**Assumption 4** (Random directions). *The set of random directions $\mathcal{D} \subset \mathbb{R}^d$ satisfies:*

1. *There exist a norm $\|\cdot\|_p, p \in [1, 2]$ and a constant $\alpha_p \in (0, 1]$, such that for all $g \in \mathbb{R}^d$:*

$$\mathbb{E}_{\mathbf{e} \in \mathcal{D}}|\langle g, \mathbf{e} \rangle| \geq \alpha_p \|g\|_p.$$

2. *For all $\mathbf{e} \in \mathcal{D}$, the norms $\|\mathbf{e}\|_2 \leq 1, \|\mathbf{e}\|_q \leq 1, \frac{1}{p} + \frac{1}{q} = 1$.*

In our paper, we consider the following instances of $\mathcal{D}$ and provide explicit constants (Bergou et al., 2020, Lemma 3.4) for them:

1. Uniform distribution on the unit Euclidean sphere $S_2^d := \{\mathbf{e}, \|\mathbf{e}\|_2 = 1\}, p = 2, \alpha_p = \frac{1}{\sqrt{2\pi d}}$. The spheres of radius $r < 1$ are feasible too.

2. Uniform distribution on standard basic vectors $\{e_1, \ldots, e_d\}, p = 1, \alpha_p = \frac{1}{d}$.

## 2.2 Convergence properties of the backbone method CompSGD

In (Lobanov et al., 2024), the authors propose a nameless procedure for the comparison oracle. For the current point $x^k$, it takes a random direction $\mathbf{e}^k$ scaled by stepsize $\gamma_k$ and goes along it where noisy function value is smaller, i.e.,

$$x^{k+1} = x^k - \gamma_k \cdot \text{sign}(f(x^k + \gamma_k \mathbf{e}^k, \xi_+) - f(x^k - \gamma_k \mathbf{e}^k, \xi_-)) \cdot \mathbf{e}^k.$$

If value $f(x^k - \gamma_k \mathbf{e}^k, \xi_-)$ is smaller than $f(x^k + \gamma_k \mathbf{e}^k, \xi_+)$, then sign equals to 1 and $x^{k+1} = x^k - \gamma_k \mathbf{e}^k$. Otherwise, the point $x^{k+1} = x^k - \gamma_k \mathbf{e}^k$ is chosen. We name it CompSGD (Alg. 1) and prove the following lemma on its convergence.

---

**Algorithm 1** CompSGD

**Input:** Starting point $x^1 \in \mathbb{R}^d$, number of iterations $T$, stepsizes $\{\gamma_k\}_{k=1}^T$;
1: **for** $k = 1, \ldots, T$ **do**
2:     Sample direction $\mathbf{e}^k$ and noise $\xi^k$ ;
3:     $\phi^k := \text{sign}\left[f(x^k + \gamma_k \mathbf{e}^k, \xi_+^k) - f(x^k - \gamma_k \mathbf{e}^k, \xi_-^k)\right]$;
4:     $x^{k+1} = x^k - \gamma_k \cdot \phi^k \cdot \mathbf{e}^k$;
5: **end for**
**Output:** uniformly random point from $\{x^1, \ldots, x^T\}$ ;

---

**Lemma 1** (CompSGD **Convergence Lemma**). *Consider lower-bounded $(L_0, L_1)$-smooth function $f$ (As. 1, 2), random directions (As. 4) and HT function estimates $\sigma_k$ (As. 3). Then Alg. 1 after $T$ iterations starting with $\Delta := f(x^1) - f^*$ and non-increasing stepsizes $\gamma_k \leq \alpha_p^2/(48 L_1 d^{\frac{1}{p} - \frac{1}{2}} \log \frac{1}{\delta})$ achieves with probability at least $1 - \delta$:*

$$\frac{\alpha_p}{8} \sum_{k=1}^T \gamma_k \|\nabla f(x^k)\|_p \leq 8\Delta + 32 L_0 \sum_{k=1}^T \gamma_k^2 + 64 \sum_{k=1}^T \tilde{\sigma}_k + \frac{48 d^{\frac{1}{p} - \frac{1}{2}}}{\alpha_p}(\gamma_1 \|\nabla f(x^1)\|_p + C_T L_0) \log(\frac{1}{\delta}). \tag{6}$$

*where $\tilde{\sigma}_k = \sigma_k$ for independent noise and $\tilde{\sigma}_k = \gamma_k \sigma_k$ for Lipschitz noise, $C_T := \max_{k \in \overline{1, T}} \gamma_k \cdot \sum_{\tau=1}^{k-1} \gamma_\tau$.*

The proof is located in Appendix A.3. Remarkably, the only effect that comes from the constant $L_1$ in bound (6) is the restriction of the maximal possible stepsize $\gamma_k \leq \alpha_p/(48 L_1 d^{\frac{1}{p} - \frac{1}{2}} \log \frac{1}{\delta})$. For this reason, in case of small $L_0$, our methods can achieve faster convergence using large stepsizes instead of decreasing ones under standard smoothness.

**Noise robustness.** CompSGD can handle heavy-tailed noise since it implicitly normalizes the finite-difference gradient approximation (3) using only function comparisons. As shown in recent works (Hübler et al., 2024; Liu & Zhou, 2024), normalization eliminates heavy tails, but only until some fixed noise level. For Lipschitz noise, the bound (6) resembles the similar convergence bounds for the normalized first-order methods and requires $\sigma_k \sim \varepsilon$. However, for independent noise, it has worse dependence on $\sigma_k$ since it is not multiplied by $\gamma_k$. Thus, in order to achieve accuracy $\varepsilon$, the noise $\sigma_k$ must not exceed $\sigma_k \sim \varepsilon^2$.

## 2.3 MajorityVote-CompSGD: robust method for comparison oracle

For comparison oracle, we cannot use popular batching to lower the noise since it requires summing the explicit function values with are not available with this oracle. Fortunately, there exists the method to aggregate the signs of differences of function estimates without its direct calculation. At this point, we propose our novel MajorityVote-CompSGD (Algorithm 2) which reduces noise level via the majority voting (Bernstein et al., 2018b) over comparison signs. Our method chooses positive or negative direction along the random vector $\mathbf{e}^k$ based on the majority of votes after several trials.

**Additional noise assumption.** The majority voting demonstrates great performance in distributed optimization (Bernstein et al., 2018b; Jin et al., 2020). In order to be effective, it must decrease probability $\mathbb{P}\left[\text{sign}(f(x^k + \gamma_k \mathbf{e}^k) - f(x^k - \gamma_k \mathbf{e}^k)) \neq \text{sign}\left[\sum_{i=1}^M \phi_i^k\right]\right]$ with the growth of $M$. However, it does not hold true for some very skewed noise distributions which sign from mean differs from mean

---

**Algorithm 2** MajorityVote-CompSGD

---

**Input:** Starting point $x^1 \in \mathbb{R}^d$, number of iterations $T$, stepsizes $\{\gamma_k\}_{k=1}^T$, batchsizes $\{M_k\}_{k=1}^T$.

1: **for** $k = 1, \dots, T$ **do**
2:     Sample direction $\mathbf{e}^k$ and noises $\{\xi_i^k\}_{i=1}^{M_k}$;
3:     $\phi_i^k = \text{sign}\left[f(x^k + \gamma_k \mathbf{e}^k, \xi_{i,+}^k) - f(x^k - \gamma_k \mathbf{e}^k, \xi_{i,-}^k)\right]$;
4:     $x^{k+1} = x^k - \gamma_k \cdot \text{sign}\left(\sum_{i=1}^{M_k} \phi_i^k\right) \cdot \mathbf{e}^k$;
5: **end for**

**Output:** uniformly random point from $\{x^1, \dots, x^T\}$.

---

of signs. Choosing the most frequent value from the sign sequence $\{\phi_i^k\}_{i=1}^M$ is actually $M$ Bernoulli trials. In these trials, the probability of choosing a correct answer grows only if the probability of failure for each $i$ less than $\frac{1}{2}$, i.e., $\mathbb{P}\left[\text{sign}(f(x^k + \gamma_k \mathbf{e}^k) - f(x^k - \gamma_k \mathbf{e}^k)) \neq \phi_i^k\right] < \frac{1}{2}, \forall i \in \overline{1, M}$. For example, this condition is satisfied if the noise for each $i$ is *unimodal and symmetric*. We use this assumption in our paper, but other assumptions from (Safaryan & Richtárik, 2021) are valid as well.

**Theorem 1** (**HP complexity for** MajorityVote-CompSGD, **independent noise**). *Consider lower-bounded $(L_0, L_1)$-smooth function $f$ (As. 1, 2), random directions with $\alpha_p$ (As. 4) and function estimates with HT **independent, unimodal and symmetric** noise $\kappa > 0$ (As. 3). Then Alg. 2 requires comparison number $N$ to achieve $\frac{1}{T}\sum_{k=1}^T \|\nabla f(x_k)\|_p \leq \varepsilon$ with probability at least $1 - \delta$ for:*

***Optimal tuning:*** $T = O\left(\frac{\Delta L_1^{\delta,p}}{\alpha_p^3 \varepsilon}\right), \gamma_k = \frac{\alpha_p^2}{48 L_1^{\delta,p}}, M_k = \max\left\{\frac{160}{\kappa^2}, \left(\frac{4\sigma T}{\Delta}\right)^2\right\}$ *for* $\varepsilon \geq \frac{4L_0}{L_1}$ *and* $T = O\left(\frac{\Delta L_0^{\delta,p}}{\alpha_p^3 \varepsilon^2}\right), \gamma_k = \sqrt{\frac{\alpha_p \Delta}{32 T L_0^{\delta,p}}}, M_k = \max\left\{\frac{160}{\kappa^2}, \left(\frac{4\sigma T}{\Delta}\right)^2\right\}$ *for* $\varepsilon \leq \frac{4L_0}{L_1}$ :

$$N = O\left(\left(\frac{\Delta L_1^{\delta,p}}{\alpha_p^3 \varepsilon} + \frac{\Delta L_0^{\delta,p}}{\alpha_p^3 \varepsilon^2}\right)\left[\frac{1}{\kappa^2} + \frac{\sigma^2}{\alpha_p^6 \varepsilon^2}\left(L_1^{\delta,p} + \frac{L_0^{\delta,p}}{\varepsilon}\right)^2\right]\right), \tag{7}$$

*where* $\Delta = f(x^1) - f^*, L_0^{\delta,p} = L_0 d^{\frac{1}{p} - \frac{1}{2}} \log(\frac{1}{\delta}), L_1^{\delta,p} = L_1 d^{\frac{1}{p} - \frac{1}{2}} \log(\frac{1}{\delta})$.

The proof is located in Appendix A.6. The results for Lipschitz noise are presented in Appendix A.7. If functions and noise parameters are unknown we propose parameter-free tuning in Theorem 4, Appendix A.6. It achieves the same dependencies on $\varepsilon$, but worsens $\Delta, L_0, L_1, \sigma$-depending factors.

**Two-stage convergence bounds.** From Theorem 1, we can clearly distinguish two phases of algorithm convergence: fast initial phase with rate $\tilde{O}\left(\varepsilon^{-3}\right)$ before threshold $\varepsilon \geq \frac{4L_0}{L_1}$ and substantially slower one with rate $\tilde{O}\left(\varepsilon^{-6}\right)$ after. In case of $L_0 \approx 0$ (e.g. for logistic regression (Gorbunov et al., 2024) and deep neural networks (Zhang et al., 2020a)), MajorityVote-CompSGD runs in the fast regime the whole time and can work with large constant stepsizes.

## 2.4 MajorityVote-CompSGD DISCUSSION

**Choice of random directions set.** Note that the choice of set $\mathcal{D}$ affects both the coefficient $\alpha_p$ and the optimized $\ell_p$-norm. For example, the Euclidean sphere, in comparison with the standard basis, has $\sqrt{d}$ times larger $\alpha_p$, but induces a smaller $\ell_2$-norm. In practice, neural networks' gradients are dense (see experiments from (Bernstein et al., 2018a)), and their norms are related by $\|\nabla f(x)\|_1 \approx \sqrt{d}\|\nabla f(x)\|_2$. Hence, the $\ell_1$-accuracy $\varepsilon'$ can be larger, around $\varepsilon' \sim \varepsilon \cdot \sqrt{d}$. Nevertheless, the Euclidean sphere is preferable for our methods, since $\alpha_p$ has more weight in our bounds.

$d$ **dependence.** For the Euclidean sphere, the bound in (7) is $\tilde{O}(\frac{d^{9/2} L_0^3}{\kappa^2 \varepsilon^6} + \frac{d^{9/2} L_1^3}{\kappa^2 \varepsilon^3})$. Compared to the prior works (§2.6), its dependence on $\varepsilon$ is standard for the BV noise. However, the $d^{9/2}$ factor is one of the largest. We emphasize that this factor only appears in high-probability bounds, which offer an additional guarantees for the solutions. Like prior works, we also prove in expectation bounds (Theorem 8, Appendix A.9), where the dependence on $d$ is considerably lower and compatible:

$$N = O\left(\left(\frac{\Delta L_1^{\delta,p}}{\alpha_p^2 \varepsilon} + \frac{\Delta L_0^{\delta,p}}{\alpha_p^2 \varepsilon^2}\right)\left[\frac{1}{\kappa^2} + \frac{\sigma^2}{\alpha_p^4 \varepsilon^2}\left(L_1^{\delta,p} + \frac{L_0^{\delta,p}}{\varepsilon}\right)^2\right]\right) \sim O\left(\frac{d^3 L_0^3}{\kappa^2 \varepsilon^6} + \frac{d^3 L_1^3}{\kappa^2 \varepsilon^3}\right).$$

**Optimality under standard smoothness.** In this paragraph, we assume that $\mathcal{D}$ is the Euclidean sphere and $L_1 = 0$. In the deterministic case $\sigma = 0$, our in expectation bound becomes $O(dL_0/\varepsilon^2)$ and exactly matches the bounds from the previous works (Bergou et al., 2020; Tang et al., 2023; Lobanov et al., 2024). Moreover, it matches the optimal bound for the deterministic zeroth-order optimization (Nemirovskij & Yudin, 1983). Our HP threshold on noise $\sigma \sim \varepsilon^2$ from (7) is the same as the threshold for the adversarial noise from OrderRCD (Lobanov et al., 2024) or for batched variance from STP with batching (Boucherouite et al., 2024).

CompSGD **proofs from (Lobanov et al., 2024).** Although CompSGD iteration is proposed in (Lobanov et al., 2024), the authors prove only asymptotic convergence with parameters depending on the solution. We demonstrate in Lemma 1 and experiments from Section 3.2 that vanilla CompSGD without noise reduction cannot achieve accuracies lower than noise $\sigma$. For this reason, we propose effective majority voting modification (Alg. 2) which converges non-asymptotically (Theorem 1).

**HP results from (Saha et al., 2021).** The noisy comparison oracle and noise reduction scheme from (Saha et al., 2021) are similar to ours. However, all results from (Saha et al., 2021) are proved for the convex functions, and we prove it for the non-convex ones. The authors used a non-trivial assumption: for some constant $\nu \in (0, 1/2)$

$$\mathbb{P}_\xi \left[ \phi(x, y, \xi) \neq \text{sign}(f(x) - f(y)) \right] \leq 1/2 - \nu, \forall x, y \in \mathbb{R}^d. \tag{8}$$

We would like to highlight that *our Assumption 3 is much weaker and general*, since (8) can fail even under BV noise as difference $f(x) - f(y)$ goes to zero. In our proofs, we show that

$$\mathbb{P}_\xi \left[ \phi(x, y, \xi) \neq \text{sign}(f(x) - f(y)) \right] \leq \sigma/|f(x) - f(y)|.$$

Thus, in the vicinity of the stationary point where the changes of the function are small or under large $\sigma$, the required condition (8) cannot hold.

### 2.5 minibatch-CompSGD: ROBUST METHOD FOR FUNCTION VALUE ORACLE

In this section, we adopt our backbone method CompSGD to zeroth-order optimization, where function values are available or one can batch function values at two points before its direct comparison (e.g. with sum-type objective function), and build minibatch-CompSGD (Algorithm 3).

---

**Algorithm 3** minibatch-CompSGD

---

**Input:** Starting point $x^1 \in \mathbb{R}^d$, number of iterations $T$, stepsizes $\{\gamma_k\}_{k=1}^T$, batchsizes $\{B_k\}_{k=1}^T$.
1: **for** $k = 1, \dots, T$ **do**
2:    Sample a random direction $\mathbf{e}^k$ and $\{\xi_{i,\pm}^k\}_{i=1}^{B_k}$;
3:    Set $x^{k+1} = x^k - \gamma_k \cdot \text{sign}(\sum_{i=1}^{B_k} f(x^k + \gamma_k \mathbf{e}^k, \xi_{i,+}^k) - \sum_{i=1}^{B_k} f(x^k - \gamma_k \mathbf{e}^k, \xi_{i,-}^k)) \cdot \mathbf{e}^k$;
4: **end for**
**Output:** uniformly random point from $\{x^1, \dots, x^T\}$.

---

**Theorem 2** (**HP complexity for** minibatch-CompSGD). *Consider lower-bounded $(L_0, L_1)$-smooth function $f$ (As. 1, 2), random directions (As. 4) and HT function estimates $\kappa \in (1, 2]$ (As. 3). Then Alg. 3 requires $N$ function calls to achieve $\frac{1}{T} \sum_{k=1}^T \|\nabla f(x_k)\|_p \leq \varepsilon$ with probability at least $1 - \delta$:*

***Optimal tuning, independent noise:*** $T = O\left(\frac{\Delta L_1^{\delta,p}}{\alpha_p^3 \varepsilon}\right), \gamma_k = \frac{\alpha_p^2}{48 L_1^{\delta,p}}, B_k = \max\left\{1, \left(\frac{4\sigma T}{\Delta}\right)^{\frac{\kappa}{\kappa-1}}\right\}$ *for*

$\varepsilon \geq \frac{4L_0}{L_1}$ *and* $T = O\left(\frac{\Delta L_0^{\delta,p}}{\alpha_p^3 \varepsilon^2}\right), \gamma_k = \sqrt{\frac{\alpha_p \Delta}{32 T L_0^{\delta,p}}}, B_k = \max\left\{1, \left(\frac{4\sigma T}{\Delta}\right)^{\frac{\kappa}{\kappa-1}}\right\}$ *for* $\varepsilon \leq \frac{4L_0}{L_1}$

$$N = O\left(\frac{\Delta}{\alpha_p^3 \varepsilon}\left(L_1^{\delta,p} + \frac{L_0^{\delta,p}}{\varepsilon}\right)\left[1 + \left(\frac{\sigma}{\alpha_p^3 \varepsilon}\left(L_1^{\delta,p} + \frac{L_0^{\delta,p}}{\varepsilon}\right)\right)^{\frac{\kappa}{\kappa-1}}\right]\right),$$

***Optimal tuning, Lipschitz noise:*** $T = O\left(\frac{\Delta L_1^{\delta,p}}{\alpha_p^3 \varepsilon}\right), \gamma_k = \frac{\alpha_p^2}{48 L_1^{\delta,p}}, B_k = \max\left\{1, \left(\frac{16\sigma}{\alpha_p \varepsilon}\right)^{\frac{\kappa}{\kappa-1}}\right\}$ *for*

$\varepsilon \geq \frac{4L_0}{L_1}$ *and* $T = O\left(\frac{\Delta L_0^{\delta,p}}{\alpha_p^3 \varepsilon^2}\right), \gamma_k = \sqrt{\frac{\alpha_p \Delta}{32 T L_0^{\delta,p}}}, B_k = \max\left\{1, \left(\frac{16\sigma}{\alpha_p \varepsilon}\right)^{\frac{\kappa}{\kappa-1}}\right\}$ *for* $\varepsilon \leq \frac{4L_0}{L_1}$

$$N = O\left(\frac{\Delta d^{\frac{1}{p} - \frac{1}{2}}}{\alpha_p^3 \varepsilon}\left(L_1 + \frac{L_0}{\varepsilon}\right)\left[1 + \left(\frac{\sigma}{\alpha_p \varepsilon}\right)^{\frac{\kappa}{\kappa-1}}\right]\right)\log 1/\delta,$$

*where* $\Delta = f(x^1) - f^*, L_0^{\delta,p} = L_0 d^{\frac{1}{p} - \frac{1}{2}} \log \frac{1}{\delta}, L_1^{\delta,p} = L_1 d^{\frac{1}{p} - \frac{1}{2}} \log \frac{1}{\delta}.$

The proof is located in Appendices A.4 and A.5. Parameter-free tuning for minibatch-CompSGD can be done only under Lipschitz noise (Theorem 3, Appendix A.4). We also provide in expectation bounds for minibatch-CompSGD with weaker $d$ dependence in (Theorem 7, Appendix A.8).

### 2.6 minibatch-CompSGD DISCUSSION

**Related works.** For a fair comparison with other methods, we use our in expectation bounds in Theorem 7 under BV noise $\kappa = 2$ and standard smoothness $L_1 = 0$, namely, minibatch-CompSGD on $\mathcal{D} = S_2^d$ achieves rates $O(d^3\varepsilon^{-6})$ and $O(d^2\varepsilon^{-4})$ for independent and Lipschitz noises. Under the same assumptions and independent noise, STP with minibatching achieves a rate $O(d^2\varepsilon^{-6})$, the same as ours. The methods with Decision-Dependent Distributions achieve rates $O(d^4\varepsilon^{-6})$ (Liu et al., 2023a) (for bounded optimized functions) and $O(d^3\varepsilon^{-6})$ (Hikima et al., 2025). When corrupting noise is Lipschitz, ZO-SignSGD achieves $O(d^2\varepsilon^{-4})$ and ZO-SGD - $O(d\varepsilon^{-4})$. To sum up, our rates have the same $\varepsilon$ dependence and are competitive in terms of $d$ factors.

**Optimality.** Under Lipschitz noise, our high-probability rates (without $d$ factors) match the optimal rates for non-convex first-order optimization (Zhang et al., 2020b) when $L_1 = 0$. For the case $L_1 \neq 0$, they match the best-known first-order bounds, namely, for SGD with normalization (Liu & Zhou, 2024). To the best of our knowledge, no lower bounds exist for generalized smoothness.

**Technical novelty.** The proof techniques in our theoretical analysis for all methods are completely different from the standard proofs for finite-difference methods. Usually zeroth-order methods take already established convergence of a first-order method and apply it to the abstract smoothed function for which gradient estimate (3) is the unbiased estimator. The chosen first-order method itself does not mean much, while the properties of the smoothed function are adjusted and proved to fit the base method. Our proof is based only on the direct structure of CompSGD method and the following HP properties. It allows us to work under much weaker assumptions, since we do not adjust to the base method. For example, the proof for ZO-SignSGD with majority voting requires the difference $[f(x + \tau\mathbf{e}, \xi) - f(x - \tau\mathbf{e}, \xi)]\mathbf{e}$ to be symmetric for both $\xi$ and $\mathbf{e}$ (what is very strict), while we only need symmetry of the noisy function estimates.

## 3 EXPERIMENTS

In this section, we present experimental results for the comparison-based and zeroth-order methods from Section 2. To demonstrate their effectiveness, we focus mainly on language (§3.1) and diffusion (§3.3) models fine-tuning. This choice is motivated by two factors: first, these tasks are known to exhibit heavy-tailed noise (Zhang et al., 2020b) and generalized smoothness (Zhang et al., 2020a; Liu et al., 2023b) characteristics, and second, they represent an important real-world application domain.

### 3.1 minibatch-CompSGD ON ROBERTA FINE-TUNING

First, we consider a zeroth-order language model fine-tuning. Following MeZO (Malladi et al., 2023a), we evaluate our method on classification fine-tuning tasks, specifically SST-2, SST-5 (Socher et al., 2013), SNLI (Bowman et al., 2015), MNLI (Williams et al., 2017) and RTE, TREC (Voorhees & Tice, 2000), on the RoBERTa-large model with $k = 16$ (Liu et al., 2019b). We employ the established few-shot prediction setting (Malladi et al., 2023b; Gao et al., 2020a). See details in Appendix B.1.

We compare minibatch-CompSGD Algorithm 3 with the pre-trained model without fine-tuning (Zero-shot) and the original MeZO version. As demonstrated in Table 1, the sign-based method outperforms its non-sign counterpart.

Table 1: Accuracy of RoBERTa-large (350M params) fine-tuned on different tasks. Higher is better.

| Method | SST-2 | SST-5 | SNLI | MNLI | RTE | TREC |
|---|---|---|---|---|---|---|
| minibatch-CompSGD | **91.9** | **46.7** | **69.6** | **63.8** | **65.7** | **77.2** |
| MeZO | 91.7 | 45.5 | 68.5 | 58.7 | 64.0 | 76.9 |
| Zero-shot | 79.0 | 35.5 | 50.2 | 48.8 | 51.4 | 32.0 |

### 3.2 MajorityVote-CompSGD FOR ACCURACY MAXIMIZATION

Second, we simulate the zeroth-order environment with comparison oracles as follows. We take the prediction accuracy of the linear model on the training dataset as the objective:

$$f(x) = \left(1 - \text{Acc}\left(\mathbf{y}_{\text{train}}, \text{sign}\left(\frac{2}{1+\exp(-\mathbf{X}_{\text{train}}x)}\right) - 1\right)\right).$$

As training data, we consider classification tasks from LibSVM (Chang & Lin, 2011): `mushrooms`, `phishing`, `a9a`. In Figure 1, we provide the dynamics of accuracy on the test dataset for our methods and for several baselines that also consider the comparison oracle: OrderRCD (Lobanov et al., 2024), STP (Bergou et al., 2020) and SMTP (Gorbunov et al., 2019). In all cases, MajotityVote-CompSGD outperforms baselines and Comp-SGD is either on par with them or better.

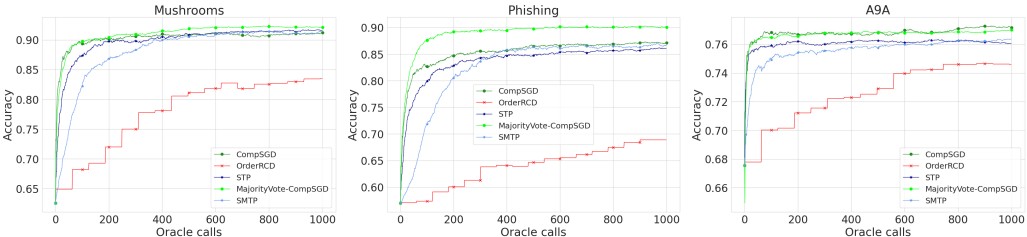

Figure 1: Performance of zeroth-order methods with comparison oracle on LibSVM datasets.

### 3.3 MajorityVote-CompSGD ON DIFFUSION MODELS FINE-TUNING

In this section, we are interested in solving the following problem: suppose that we have an image generative model; our goal is to add to its output new complex features/attributes through fine-tuning using only a comparison oracle. This setting covers a scenario in which a person can customize a model simply by selecting a preferred image. In our experiments, we simulate this human feedback with the feedback from Gemini-2.0-flash (Team, 2025).

The basic model considered is pre-trained Stable Diffusion v2.0 (Rombach et al., 2022) (dreamlike-art/dreamlike-photoreal-2.0). Since we use the model pre-trained for generating photorealistic faces, we consider freckles and green eyes as the target features. The validation is done in the following way: we generate 100 images with pre-trained model and 100 images with fine-tuned model and ask LLM-as-a-judge (Zheng et al., 2023) to score each set. To properly score the generated images, we use the two-metric scoring system: basic metrics and draw resolution. That is, image score is based on the explicit presence of target attributes, and priority is given to images with neutral backgrounds or without artificial colour distortion (see the corresponding prompts in Appendix C).

We fine-tune U-Net of Stable Diffusion using MajorityVote-CompSGD for $T = 80$ iterations with $M_k = 3$ and cosine annealing learning rate schedule with $\gamma_{\max} = 0.05$. We choose between two generated images using Gemini-2.0-flash (see the corresponding prompt in Appendix C).

As a result, we observe that pre-trained model achieves $32/100$ notional units, while our fine-tuned model scores $84/100$, indicating that even with that simple procedure it is possible to add significant features to the models output. We also provide corresponding images in Figure 2.

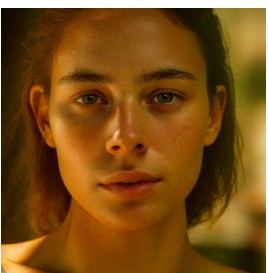 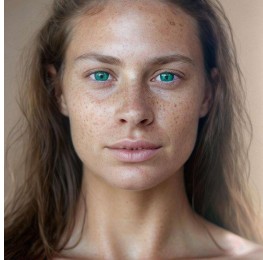

Figure 2: The images generated by the pre-trained model (left) and fine-tuned model (right).

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

## CONTENTS

# A  THEORETICAL DETAILS AND PROOFS

## A.1  $(L_0, L_1)$-SMOOTHNESS

Standard $L$-smoothness assumes that the gradient of a function is globally Lipschitz continuous. However, this condition can be too restrictive in practice. Many functions arising in optimization, especially in Machine Learning and statistics, either do not satisfy $L$-smoothness or satisfy it with a very large constant $L_0$, leading to overly pessimistic theoretical guarantees. $(L_0, L_1)$-smoothness (Assumption 2) is weaker than $L$-smoothness and allows finer control over the smoothness behavior of functions with rapidly growing curvature in regions where the gradient is large.

Importantly, many functions satisfy $(L_0, L_1)$-smoothness with *significantly smaller constants* $L_0$ and $L_1$ compared to the $L$ required for global Lipschitz smoothness. As a result, optimization algorithms tailored for $(L_0, L_1)$-smooth functions can achieve better convergence guarantees, especially in settings involving large gradients or heavy-tailed noise. The examples of practically used $(L_0, L_1)$-smooth functions include:

**Example 1** (Power of Norm). *Let $f(x) = \|x\|^{2n}$, where $n$ is a positive integer. Then, $f(x)$ is convex and $(2n, 2n-1)$-smooth. Moreover, $f(x)$ is not $L$-smooth for $n \geq 2$ and any $L \geq 0$.*

**Example 2** (Exponent of the Inner Product). *Function $f(x) = \exp(a^\top x)$ for some $a \in \mathbb{R}^d$ is convex, $(0, \|a\|)$-smooth, but not $L$-smooth for $a \neq 0$ and any $L \geq 0$.*

**Example 3** (Logistic Function). *Consider logistic function: $f(x) = \log\left(1 + \exp(-a^\top x)\right)$, where $a \in \mathbb{R}^d$ is some vector. It is known that this function is $L$-smooth and convex with $L = \|a\|^2$. However, one can show that $f$ is also $(L_0, L_1)$-smooth with $L_0 = 0$ and $L_1 = \|a\|$. For $\|a\| \gg 1$, both $L_0$ and $L_1$ are much smaller than $L$.*

**Example 4** (Quadratic Function with Linear Term.). *Let $f(x) = \frac{1}{2}x^\top A x + b^\top x$, where $A \in \mathbb{R}^{d \times d}$ is symmetric positive semi-definite, and $b \in \mathbb{R}^d$. Then $f$ is convex and $(L_0, 0)$-smooth with $L_0 = \|A\|$. This function is also $L$-smooth with the same $L$, but here $(L_1 = 0)$ shows the gradient is Lipschitz regardless of gradient size.*

The condition of $(L_0, L_1)$-smoothness from Assumption 2 can be formulated in terms of inequalities without sup operator, similar to the case of standard smoothness.

**Lemma 2** (($L_0, L_1$)-Smoothness properties. (Gorbunov et al., 2024)). *For $(L_0, L_1)$-smooth function $f$ and $x, y \in \mathbb{R}^d$, it holds*

$$\|\nabla f(x) - \nabla f(y)\|_2 \leq (L_0 + L_1\|\nabla f(y)\|_2)\exp(L_1\|x - y\|_2)\|x - y\|_2,$$

$$f(y) - f(x) - \langle \nabla f(x), y - x \rangle \leq \frac{L_0 + L_1\|\nabla f(x)\|_2}{2}\exp(L_1\|x - y\|_2)\|x - y\|_2^2. \tag{9}$$

## A.2  TECHNICAL LEMMAS AND PROPOSITIONS

We use the following facts from the linear algebra and convex analysis (Boyd, 2004):

**Proposition 1** (Norm Relation). *For two norms $\ell_p$ and $\ell_q$ with $1 \leq p \leq q \leq 2$, the following relation holds true:*

$$\|x\|_q \leq \|x\|_p \leq d^{\frac{1}{p} - \frac{1}{q}}\|x\|_q, \quad \forall x \in \mathbb{R}^d. \tag{10}$$

**Proposition 2** (Jensen's Inequality). *For scalar random variable $\xi$ with bounded $\kappa$-th moment $\kappa \in (1, 2]$, the following inequality holds true:*

$$\mathbb{E}[|\xi|] \leq (\mathbb{E}[|\xi|^\kappa])^{\frac{1}{\kappa}}. \tag{11}$$

**Proposition 3** (Markov's Inequality). *For scalar random variable $\xi$ with bounded first moment, the following inequality holds true for any $a > 0$:*

$$\mathbb{P}(|\xi - \mathbb{E}[\xi]| \geq a) \leq \frac{\mathbb{E}[|\xi|]}{a}. \tag{12}$$

To prove the HP bounds with the logarithmic dependence, we use the following measure concentration result (see, for example, (Li & Orabona, 2020, Lemma 1).

**Lemma 3** (Measure Concentration Lemma). *Let $\{D_k\}_{k=1}^T$ be a martingale difference sequence (MDS), i.e., $\mathbb{E}[D_k|D_{k-1},\ldots,D_1] = 0$ for all $k \in \overline{1,T}$. Furthermore, for each $k \in \overline{1,T}$, there exists positive $\sigma_k \in \mathbb{R}$, s.t. $\mathbb{E}\left[\exp\left(\frac{D_k^2}{\sigma_k^2}\right)|k\right] \leq e$. Then the following probability bound holds true:*

$$\forall \lambda > 0, \delta \in (0,1): \quad \mathbb{P}\left(\sum_{k=1}^T D_k \leq \frac{3}{4}\lambda \sum_{k=1}^T \sigma_k^2 + \frac{1}{\lambda}\log(1/\delta)\right) \geq 1 - \delta. \tag{13}$$

To control error reduction during batching, we use the following batching lemma for HT variables. Its modern proof for $d = 1$ was proposed in (Cherapanamjeri et al., 2022, Lemma 4.2) and then generalized for the multidimensional case in (Kornilov et al., 2024; Hübler et al., 2024).

**Lemma 4** (HT Batching Lemma). *Let $\kappa \in (1,2]$, and $X_1,\ldots,X_B \in \mathbb{R}^d$ be a martingale difference sequence (MDS), i.e., $\mathbb{E}[X_i|X_{i-1},\ldots,X_1] = 0$ for all $i \in \overline{1,B}$. If all variables $X_i$ have bounded $\kappa-$th moment, i.e., $\mathbb{E}[\|X_i\|_2^\kappa] < +\infty$, then the following bound holds true*

$$\mathbb{E}\left[\left\|\frac{1}{B}\sum_{i=1}^B X_i\right\|_2^\kappa\right] \leq \frac{2}{B^\kappa}\sum_{i=1}^B \mathbb{E}[\|X_i\|_2^\kappa]. \tag{14}$$

## A.3 PROOF OF CompSGD CONVERGENCE LEMMA 1

*Proof.* Consider the $k$-th step of CompSGD. We use smoothness of function $f$ (Lemma 2) to estimate:

$$
\begin{aligned}
f(x^{k+1}) - f(x^k) &\leq & \langle \nabla f(x^k), x^{k+1} - x^k \rangle + \frac{L_0 + L_1\|\nabla f(x^k)\|_2}{2}\exp(L_1\|x^{k+1} - x^k\|_2)\|x^{k+1} - x^k\|_2^2 \\
&=& -\gamma_k \cdot \mathrm{sign}(f(x^k + \gamma_k \mathbf{e}^k, \xi_+^k) - f(x^k - \gamma_k \mathbf{e}^k, \xi_-^k)) \cdot \langle \nabla f(x^k), \mathbf{e}^k \rangle \\
&& + \frac{L_0 + L_1\|\nabla f(x^k)\|_2}{2}\exp(\gamma_k L_1\|\mathbf{e}^k\|_2)\gamma_k^2\|\mathbf{e}^k\|_2^2 \\
&\overset{As.4}{\leq}& -\gamma_k \frac{\mathrm{sign}(f(x^k + \gamma_k \mathbf{e}^k, \xi_+^k) - f(x^k - \gamma_k \mathbf{e}^k, \xi_-^k)) \cdot \langle \nabla f(x^k), \mathbf{e}^k \rangle}{\|\nabla f(x^k)\|_p} \cdot \|\nabla f(x^k)\|_p \\
&& + \frac{L_0 + L_1\|\nabla f(x^k)\|_2}{2}\exp(\gamma_k L_1)\gamma_k^2.
\end{aligned}
$$

Let us choose $\gamma_k \leq \frac{1}{4L_1} \leq \frac{\alpha_p}{4L_1}$, then we have $L_1\gamma_k \exp(L_1\gamma_k) \leq \frac{\alpha_p}{4}$ and $\|\nabla f(x^k)\|_2 \leq \|\nabla f(x^k)\|_p$:

$$
\begin{aligned}
f(x^{k+1}) - f(x^k) &\leq& -\gamma_k \frac{\mathrm{sign}(f(x^k + \gamma_k \mathbf{e}^k, \xi_+^k) - f(x^k - \gamma_k \mathbf{e}^k, \xi_-^k)) \cdot \langle \nabla f(x^k), \mathbf{e}^k \rangle}{\|\nabla f(x^k)\|_p} \cdot \|\nabla f(x^k)\|_p \\
&& + \frac{L_0\gamma_k^2}{2} + \frac{\alpha_p\gamma_k\|\nabla f(x^k)\|_p}{8}. \tag{15}
\end{aligned}
$$

Consequently, after summing $T$ steps, we obtain

$$\sum_{k=1}^T \gamma_k \left[\frac{\mathrm{sign}(f(x^k + \gamma_k \mathbf{e}^k, \xi_+^k) - f(x^k - \gamma_k \mathbf{e}^k, \xi_-^k)) \cdot \langle \nabla f(x^k), \mathbf{e}^k \rangle}{\|\nabla f(x^k)\|_p} - \frac{\alpha_p}{8}\right] \cdot \|\nabla f(x^k)\|_p \leq$$

$$\underbrace{f(x^1) - f(x^*)}_{=\Delta} + \frac{L_0}{2}\sum_{k=1}^T \gamma_k^2. \tag{16}$$

Next, we deal with terms $\phi_k := \frac{\mathrm{sign}(f(x^k+\gamma_k \mathbf{e}^k, \xi_+^k) - f(x^k-\gamma_k \mathbf{e}^k, \xi_-^k)) \cdot \langle \nabla f(x^k), \mathbf{e}^k \rangle}{\|\nabla f(x^k)\|_p}$, $\psi_k := \mathbb{E}[\phi_k|x^k]$ and $D_k := -\gamma_k(\phi_k - \psi_k)\|\nabla f(x^k)\|_p$. The terms $\phi_k$ are bounded with $|\phi_k| \leq 1$ due to Cauchy–Schwarz inequality :

$$|\phi_k| = \frac{|\mathrm{sign}(f(x^k + \gamma_k \mathbf{e}^k, \xi_+^k) - f(x^k - \gamma_k \mathbf{e}^k, \xi_-^k)) \cdot \langle \nabla f(x^k), \mathbf{e}^k \rangle|}{\|\nabla f(x^k)\|_p} \leq \frac{|\langle \nabla f(x^k), \mathbf{e}^k \rangle|}{\|\nabla f(x^k)\|_p} \leq \|\mathbf{e}^k\|_q \overset{As.4}{\leq} 1.$$

We note that $D_k$ is a martingale difference sequence ($\mathbb{E}[D_k|D_{k-1}, \ldots, D_1] = 0$) satisfying the inequality

$$\exp\left(\frac{D_k^2}{4\gamma_k^2\|\nabla f(x^k)\|_p^2}\right) = \exp\left(\frac{(\phi_k - \psi_k)^2}{4}\right) \leq e.$$

Applying Measure Concentration Lemma 3 to the sequence $D_k$ with $\sigma_k^2 = 4\gamma_k^2\|\nabla f(x^k)\|_p^2$, we derive the bound for all $\lambda > 0$ with probability at least $1 - \delta$ :

$$\sum_{k=1}^{T} \gamma_k \left(\psi_k - 3\lambda\gamma_k\|\nabla f(x^k)\|_p - \frac{\alpha_p}{8}\right)\|\nabla f(x^k)\|_p \leq \Delta + \frac{L_0}{2}\sum_{k=1}^{T}\gamma_k^2 + \frac{1}{\lambda}\log(1/\delta)$$

We use norm relation (10) and $(L_0, L_1)$-smoothness (Lemma 2) to estimate maximum gradient norm for all $k \in \overline{2, T+1}$ :

$$
\begin{aligned}
\|\nabla f(x^k)\|_p/d^{\frac{1}{p}-\frac{1}{2}} &\leq \|\nabla f(x^k)\|_2 \leq \|\nabla f(x^k) - \nabla f(x^{k-1}) + \nabla f(x^{k-1})\|_2 \\
&\leq \|\nabla f(x^k) - \nabla f(x^{k-1})\|_2 + \|\nabla f(x^{k-1})\|_2 \\
&\leq (L_0 + L_1\|\nabla f(x^{k-1})\|_2)\exp(L_1\|x^k - x^{k-1}\|_2)\|x^k - x^{k-1}\|_2 + \|\nabla f(x^{k-1})\|_2 \\
&\leq (L_0 + L_1\|\nabla f(x^{k-1})\|_2)\exp(L_1\gamma_k)\gamma_k + \|\nabla f(x^{k-1})\|_2.
\end{aligned}
$$

At this point, we take $\gamma_k \leq \frac{\alpha_p^2}{48 L_1 d^{\frac{1}{p}-\frac{1}{2}}\log\frac{1}{\delta}}$ to obtain

$$
\begin{aligned}
\|\nabla f(x^k)\|_p/d^{\frac{1}{p}-\frac{1}{2}} &\leq 2L_0\gamma_k + \frac{\alpha_p^2\|\nabla f(x^{k-1})\|_2}{48d^{\frac{1}{p}-\frac{1}{2}}\log\frac{1}{\delta}} + \|\nabla f(x^{k-1})\|_2 \\
&\leq 2L_0\sum_{\tau=1}^{k-1}\gamma_\tau + \sum_{\tau=1}^{k-1}\frac{\alpha_p^2\|\nabla f(x^\tau)\|_2}{48d^{\frac{1}{p}-\frac{1}{2}}\log\frac{1}{\delta}} + \|\nabla f(x^1)\|_2 \\
&\leq 2L_0\sum_{\tau=1}^{k-1}\gamma_\tau + \sum_{\tau=1}^{k-1}\frac{\alpha_p^2\|\nabla f(x^\tau)\|_p}{48d^{\frac{1}{p}-\frac{1}{2}}\log\frac{1}{\delta}} + \|\nabla f(x^1)\|_p,
\end{aligned}
$$

$$\gamma_k\|\nabla f(x^k)\|_p \leq 2L_0 d^{\frac{1}{p}-\frac{1}{2}}\cdot\gamma_k\sum_{\tau=1}^{k-1}\gamma_\tau + \gamma_k\sum_{\tau=1}^{k-1}\frac{\alpha_p^2\|\nabla f(x^\tau)\|_p}{48\log\frac{1}{\delta}} + d^{\frac{1}{p}-\frac{1}{2}}\gamma_k\|\nabla f(x^1)\|_p.$$

Since stepsizes $\gamma_k$ are non-increasing, we have

$$\gamma_k\sum_{\tau=1}^{k-1}\frac{\alpha_p^2\|\nabla f(x^\tau)\|_p}{48\log\frac{1}{\delta}} \leq \sum_{\tau=1}^{k-1}\frac{\gamma_\tau\alpha_p^2\|\nabla f(x^\tau)\|_p}{48\log\frac{1}{\delta}},$$

$$\gamma_k\|\nabla f(x^k)\|_p \leq 2L_0 d^{\frac{1}{p}-\frac{1}{2}}\cdot\gamma_k\sum_{\tau=1}^{k-1}\gamma_\tau + \sum_{\tau=1}^{k-1}\frac{\gamma_\tau\alpha_p^2\|\nabla f(x^\tau)\|_p}{48\log\frac{1}{\delta}} + d^{\frac{1}{p}-\frac{1}{2}}\gamma_k\|\nabla f(x^1)\|_p.$$

Hence, the choice $\lambda := \frac{\alpha_p}{6d^{\frac{1}{p}-\frac{1}{2}}\left(\gamma_1\|\nabla f(x^1)\|_p + \sum_{\tau=1}^{k-1}\frac{\alpha_p^2\gamma_\tau\|\nabla f(x^\tau)\|_p}{48d^{\frac{1}{p}-\frac{1}{2}}\log\frac{1}{\delta}} + C_T L_0\right)}$ yields with probability at least $1 - \delta$:

$$\sum_{k=1}^{T}\gamma_k\left(\psi_k - \frac{\alpha_p}{2} - \frac{\alpha_p}{8} - \frac{\alpha_p}{8}\right)\|\nabla f(x^k)\|_p \leq \Delta + \frac{L_0}{2}\sum_{k=1}^{T}\gamma_k^2 + \frac{6d^{\frac{1}{p}-\frac{1}{2}}}{\alpha_p}(\gamma_1\|\nabla f(x^1)\|_2 + C_T L_0)\log(1/\delta), \quad (17)$$

where $C_T := \max_{k\in\overline{1,T}}\gamma_k\cdot\sum_{\tau=1}^{k-1}\gamma_\tau$ and $\gamma_1 := \max_{k\in\overline{1,T}}\gamma_k$. Next, we estimate the term $\psi_k\|\nabla f(x^k)\|_p$:

$$\mathbb{E}_{\xi,\mathbf{e}^k}\left[\text{sign}(f(x^k + \gamma_k\mathbf{e}^k, \xi_+^k) - f(x^k - \gamma_k\mathbf{e}^k, \xi_-^k))\cdot\langle\nabla f(x^k), \mathbf{e}^k\rangle\right] = \mathbb{E}_{\mathbf{e}^k}|\langle\nabla f(x^k), \mathbf{e}^k\rangle|$$

$$- \mathbb{E}_{\mathbf{e}^k}\left[2\cdot\mathbb{P}_\xi\left[\text{sign}(f(x^k + \gamma_k\mathbf{e}^k, \xi_+^k) - f(x^k - \gamma_k\mathbf{e}^k, \xi_-^k)) \neq \text{sign}(\langle\nabla f(x^k), \mathbf{e}^k\rangle)\right]\cdot|\langle\nabla f(x^k), \mathbf{e}^k\rangle|\right].$$

We consider two cases to deal with probability over $\xi$: $|\langle\nabla f(x^k), \mathbf{e}^k\rangle| \geq 2\gamma_k L_0 + \alpha_p\|\nabla f(x^k)\|_p/8$ and $|\langle\nabla f(x^k), \mathbf{e}^k\rangle| \leq 2\gamma_k L_0 + \alpha_p\|\nabla f(x^k)\|_p/8$.

**Case** $|\langle \nabla f(x^k), \mathbf{e}^k \rangle| \leq 2\gamma_k L_0 + \alpha_p \|\nabla f(x^k)\|_p/8$**:**

$$\mathbb{E}_{\xi, \mathbf{e}^k} \left[ \text{sign}(f(x^k + \gamma_k \mathbf{e}^k, \xi_+^k) - f(x^k - \gamma_k \mathbf{e}^k, \xi_-^k)) \cdot \langle \nabla f(x^k), \mathbf{e}^k \rangle \right] \geq -\mathbb{E}_{\mathbf{e}^k}[|\langle \nabla f(x^k), \mathbf{e}^k \rangle|]$$

$$\geq \quad \mathbb{E}_{\mathbf{e}^k}[|\langle \nabla f(x^k), \mathbf{e}^k \rangle|] - 4\gamma_k L_0 - \alpha_p \frac{\|\nabla f(x^k)\|_p}{8} \overset{\text{As. 4}}{\geq} \frac{7\alpha_p}{8} \|\nabla f(x^k)\|_p - 4\gamma_k L_0.$$

**Case** $|\langle \nabla f(x^k), \mathbf{e}^k \rangle| \geq 2\gamma_k L_0 + \alpha_p \|\nabla f(x^k)\|_p/8$**:**

We change sign operators to equivalent ones denoting $\theta_+^k := f(x^k + \gamma_k \mathbf{e}^k, \xi_+^k) - f(x^k + \gamma_k \mathbf{e}^k)$ and $\theta_-^k := f(x^k - \gamma_k \mathbf{e}^k, \xi_-^k) - f(x^k - \gamma_k \mathbf{e}^k)$:

$$\text{sign}(f(x^k + \gamma_k \mathbf{e}^k, \xi_+^k) - f(x^k - \gamma_k \mathbf{e}^k, \xi_-^k)) \quad \neq \quad \text{sign}(\langle \nabla f(x^k), \mathbf{e}^k \rangle)$$
$$\Updownarrow$$
$$\text{sign}(f(x^k + \gamma_k \mathbf{e}^k) - f(x^k - \gamma_k \mathbf{e}^k) + \theta_+^k - \theta_-^k) \quad \neq \quad \text{sign}(2\gamma_k \cdot \langle \nabla f(x^k), \mathbf{e}^k \rangle).$$

Further, we can bound probability by considering larger number of cases:

$$\mathbb{P}_\xi \left[ \text{sign}(f(x^k + \gamma_k \mathbf{e}^k, \xi_+^k) - f(x^k - \gamma_k \mathbf{e}^k, \xi_-^k)) \neq \text{sign}(\langle \nabla f(x^k), \mathbf{e}^k \rangle) \right] \tag{18}$$

$$= \quad \mathbb{P}_\xi \left[ \text{sign}(f(x^k + \gamma_k \mathbf{e}^k) - f(x^k - \gamma_k \mathbf{e}^k) + \theta_+^k - \theta_-^k) \neq \text{sign}(2\gamma_k \cdot \langle \nabla f(x^k), \mathbf{e}^k \rangle) \right]$$

$$\leq \quad \mathbb{P}_\xi \left[ |f(x^k + \gamma_k \mathbf{e}^k) - f(x^k - \gamma_k \mathbf{e}^k) + \theta_+^k - \theta_-^k - 2\gamma_k \cdot \langle \nabla f(x^k), \mathbf{e}^k \rangle| \geq 2\gamma_k \cdot |\langle \nabla f(x^k), \mathbf{e}^k \rangle| \right]$$

$$\leq \quad \mathbb{P}_\xi \left[ |f(x^k + \gamma_k \mathbf{e}^k) - f(x^k - \gamma_k \mathbf{e}^k) - 2\gamma_k \cdot \langle \nabla f(x^k), \mathbf{e}^k \rangle| + |\theta_+^k - \theta_-^k| \geq 2\gamma_k \cdot |\langle \nabla f(x^k), \mathbf{e}^k \rangle| \right].$$

We apply Smoothness Lemma 2 and choose $\gamma_k \leq \frac{\alpha_p}{8L_1}$ to bound the term:

$$|f(x^k + \gamma_k \mathbf{e}^k) - f(x^k - \gamma_k \mathbf{e}^k) - 2\gamma_k \cdot \langle \nabla f(x^k), \mathbf{e}^k \rangle| \quad \leq \quad 2 \cdot \frac{L_0 + L_1 \|\nabla f(x^k)\|_2}{2} \exp(L_1 \gamma_k \|\mathbf{e}^k\|_2) \gamma_k^2 \|\mathbf{e}^k\|_2^2$$

$$\leq \quad 2L_0 \gamma_k^2 + \alpha_p \|\nabla f(x^k)\|_p \gamma_k / 8.$$

We continue to estimate the probability:

$$\mathbb{P}_\xi \left[ |f(x^k + \gamma_k \mathbf{e}^k) - f(x^k - \gamma_k \mathbf{e}^k) - 2\gamma_k \cdot \langle \nabla f(x^k), \mathbf{e}^k \rangle| + |\theta_+^k - \theta_-^k| \geq 2\gamma_k \cdot |\langle \nabla f(x^k), \mathbf{e}^k \rangle| \right]$$

$$\leq \quad \mathbb{P}_\xi \left[ 2L_0 \gamma_k^2 + \gamma_k \alpha_p \|\nabla f(x^k)\|_p / 4 + |\theta_+^k - \theta_-^k| \geq 2\gamma_k \cdot |\langle \nabla f(x^k), \mathbf{e}^k \rangle| \right]. \tag{19}$$

Since we consider the case $|\langle \nabla f(x^k), \mathbf{e}^k \rangle| \geq 2\gamma_k L_0 + \alpha_p \|\nabla f(x^k)\|_p / 8$, we bound

$$(19) \quad \leq \quad \mathbb{P}_\xi \left[ \gamma_k \cdot |\langle \nabla f(x^k), \mathbf{e}^k \rangle| + |\theta_+^k - \theta_-^k| \geq 2\gamma_k \cdot |\langle \nabla f(x^k), \mathbf{e}^k \rangle| \right]$$

$$\leq \quad \mathbb{P}_\xi \left[ |\theta_+^k - \theta_-^k| \geq \gamma_k \cdot |\langle \nabla f(x^k), \mathbf{e}^k \rangle| \right]$$

$$\overset{\text{Markov ineq.(12):}}{\leq} \quad \frac{\mathbb{E}_\xi[|\theta_+^k - \theta_-^k|]}{\gamma_k \cdot |\langle \nabla f(x^k), \mathbf{e}^k \rangle|}. \tag{20}$$

In case of independent noise, $\mathbb{E}_\xi[|\theta_+^k - \theta_-^k|]$ is simply bounded by the constant $\sigma_k$ and $\tilde{\sigma}_k = \sigma_k$. In case of Lipschitz noise, $\mathbb{E}_\xi[|\theta_+^k - \theta_-^k|] \leq \sigma_k \|2\gamma_k \mathbf{e}^k\|_2 = 2\sigma_k \gamma_k$ and $\tilde{\sigma}_k = 2\sigma_k \gamma_k$. Finally, we obtain the bound

$$\mathbb{E}_{\xi, \mathbf{e}^k} \left[ \text{sign}(f(x^k + \gamma_k \mathbf{e}^k, \xi_+^k) - f(x^k - \gamma_k \mathbf{e}^k, \xi_-^k)) \cdot \langle \nabla f(x^k), \mathbf{e}^k \rangle \right] \quad \geq \quad \mathbb{E}_{\mathbf{e}^k} |\langle \nabla f(x^k), \mathbf{e}^k \rangle| - \frac{4\tilde{\sigma}_k}{\gamma_k}$$

$$\overset{\text{As. 4}}{\geq} \quad \alpha_p \|\nabla f(x^k)\|_p - \frac{4\tilde{\sigma}_k}{\gamma_k}.$$

Combining two cases together, we get that $\psi_k \|\nabla f(x^k)\|_p \geq \frac{7\alpha_p}{8} \|\nabla f(x^k)\|_p - 4\gamma_k L_0 - \frac{4\tilde{\sigma}_k}{\gamma_k}$, and the bound follows from (17)

$$\frac{1}{8} \sum_{k=1}^T \gamma_k \|\nabla f(x^k)\|_p \quad \leq \quad \frac{\Delta}{\alpha_p} + \frac{L_0}{2\alpha_p} \sum_{k=1}^T \gamma_k^2 + \sum_{k=1}^T \gamma_k \cdot \frac{4L_0 \gamma_k}{\alpha_p} + 4 \sum_{k=1}^T \frac{\tilde{\sigma}_k}{\alpha_p}$$

$$+ \quad \frac{6 d^{\frac{1}{p} - \frac{1}{2}}}{\alpha_p^2} (\gamma_1 \|\nabla f(x^1)\|_p + C_T L_0) \log(1/\delta).$$

Thus, we obtain the desired bound:

$$\frac{\alpha_p}{8} \sum_{k=1}^T \gamma_k \|\nabla f(x^k)\|_p \quad \leq \quad \Delta + 8L_0 \sum_{k=1}^T \gamma_k^2 + 4 \sum_{k=1}^T \tilde{\sigma}_k + \frac{6 d^{\frac{1}{p} - \frac{1}{2}}}{\alpha_p} (\gamma_1 \|\nabla f(x^1)\|_p + C_T L_0) \log(\frac{1}{\delta}).$$

$$\square$$

### A.4 PROOF OF minibatch-CompSGD COMPLEXITY THEOREM 2, INDEPENDENT NOISE

*Proof.* Due to Batching Lemma 4, we can estimate the $\kappa-$th moment of the batched estimate by:

$$\sigma_k \leq \frac{2\sigma}{B_k^{\frac{\kappa-1}{\kappa}}}.$$

We start with CompSGD Convergence Lemma 1. Plugging in constant stepsizes and batchsizes $\gamma_k \equiv \gamma, C_T = T\gamma^2, \gamma_1 = \gamma$ and dividing both sides by $\frac{\alpha_p T \gamma}{8}$ yields the bound:

$$\frac{1}{T}\sum_{k=1}^{T}\|\nabla f(x^k)\|_p \leq \frac{8\Delta}{T\alpha_p\gamma} + 64d^{\frac{1}{p}-\frac{1}{2}}\frac{L_0\gamma}{\alpha_p^2}\log(1/\delta) + \frac{32\sigma}{\alpha_p\gamma B^{\frac{\kappa-1}{\kappa}}} + \frac{48d^{\frac{1}{p}-\frac{1}{2}}\|\nabla f(x^1)\|_2}{T\alpha_p^2}\log(\frac{1}{\delta}). \tag{21}$$

The $T$ dependence in the first three terms is dominant in comparison with the last term, hence, we neglect it. Next, we pick optimal parameters for optimal convergence and prove convergence bounds for parameter-free tuning.

**Optimal tuning, $\varepsilon \geq \frac{4L_0}{L_1}$:** Choosing the largest possible $\gamma = \frac{\alpha_p^2}{48d^{\frac{1}{p}-\frac{1}{2}}L_1^\delta}$, we bound the term:

$$64d^{\frac{1}{p}-\frac{1}{2}}\frac{L_0^\delta\gamma}{\alpha_p^2} \leq \frac{64}{48}\frac{L_0}{L_1} \leq \frac{\varepsilon}{2}.$$

The bound (21) becomes:

$$\frac{1}{T}\sum_{k=1}^{T}\|\nabla f(x^k)\|_p \leq \frac{512L_1^\delta d^{\frac{1}{p}-\frac{1}{2}}}{\alpha_p^3}\left[\frac{\Delta}{T} + \frac{4\sigma}{B^{\frac{\kappa-1}{\kappa}}}\right] + \frac{\varepsilon}{2}.$$

Choosing $B$ such that $\frac{4\sigma}{\Delta B^{\frac{\kappa-1}{\kappa}}} \leq \frac{1}{T} \Rightarrow B = \max\left\{1, \left(\frac{4\sigma T}{\Delta}\right)^{\frac{\kappa}{\kappa-1}}\right\}$, we only need to bound

$$\frac{1024L_1^\delta d^{\frac{1}{p}-\frac{1}{2}}}{\alpha_p^3}\frac{\Delta}{T} \leq \frac{\varepsilon}{2} \Rightarrow T = O\left(\frac{\Delta L_1^\delta d^{\frac{1}{p}-\frac{1}{2}}}{\alpha_p^3\varepsilon}\right).$$

The final sample complexity is

$$N = T \cdot B = O\left(\frac{\Delta L_1^\delta d^{\frac{1}{p}-\frac{1}{2}}}{\alpha_p^3\varepsilon}\left[1 + \left(\frac{\sigma L_1^\delta d^{\frac{1}{p}-\frac{1}{2}}}{\alpha_p^3\varepsilon}\right)^{\frac{\kappa}{\kappa-1}}\right]\right).$$

**Optimal tuning, $\varepsilon \leq \frac{4L_0}{L_1}$:** Choosing $B$ such that $\frac{4\sigma}{\Delta B^{\frac{\kappa-1}{\kappa}}} \leq \frac{1}{T} \Rightarrow B = \max\left\{1, \left(\frac{4\sigma T}{\Delta}\right)^{\frac{\kappa}{\kappa-1}}\right\}$, we transform the bound (21) into

$$\frac{1}{T}\sum_{k=1}^{T}\|\nabla f(x^k)\|_p \leq \frac{2\Delta}{T\alpha_p\gamma} + 64d^{\frac{1}{p}-\frac{1}{2}}\frac{L_0^\delta\gamma}{\alpha_p^2} \leq \varepsilon.$$

Using $\gamma = \sqrt{\frac{\Delta\alpha_p}{32TL_0^\delta d^{\frac{1}{p}-\frac{1}{2}}}}$, we obtain

$$\frac{2\Delta}{T\alpha_p\gamma} + 64d^{\frac{1}{p}-\frac{1}{2}}\frac{L_0^\delta\gamma}{\alpha_p^2} = 16\sqrt{\frac{\Delta L_0^\delta d^{\frac{1}{p}-\frac{1}{2}}}{\alpha_p^3 T}} \leq \varepsilon.$$

Hence, required number of iterations $T = O\left(\frac{\Delta L_0^\delta d^{\frac{1}{p}-\frac{1}{2}}}{\alpha_p^3\varepsilon^2}\right)$ and the final comparison complexity is

$$N = T \cdot B = O\left(\frac{\Delta L_0^\delta d^{\frac{1}{p}-\frac{1}{2}}}{\alpha_p^3\varepsilon^2}\left[1 + \left(\frac{\sigma L_0^\delta d^{\frac{1}{p}-\frac{1}{2}}}{\alpha_p^3\varepsilon^2}\right)^{\frac{\kappa}{\kappa-1}}\right]\right).$$

We also notice that $\gamma = \sqrt{\frac{\Delta}{32TL_0^\delta d^{\frac{1}{p}-\frac{1}{2}}}} \leq \frac{\alpha_p}{48d^{\frac{1}{p}-\frac{1}{2}}L_1^\delta}$ for this number of iterations and $\varepsilon \leq \frac{4L_0}{L_1}$. $\square$

### A.5 PROOF OF minibatch-CompSGD COMPLEXITY THEOREM 2, LIPSCHITZ NOISE

The proof of Theorem 2 in case of Lipschitz noise is divided into two parts: for finite horizon with optimal tuning below and for infinite horizon with parameter-free tuning (Theorem 3).

*Proof.* Due to Batching Lemma 4, we can estimate the $\kappa-$th moment of the batched estimate by:

$$\sigma_k \leq \frac{2\sigma}{B_k^{\frac{\kappa-1}{\kappa}}}.$$

We start with CompSGD Convergence Lemma 1. Plugging in constant stepsizes and batchsizes $\gamma_k \equiv \gamma, C_T = T\gamma^2, \gamma_1 = \gamma$ and dividing both sides by $\frac{\alpha_p T\gamma}{8}$ yields the bound:

$$\frac{1}{T}\sum_{k=1}^{T}\|\nabla f(x^k)\|_p \leq \frac{8\Delta}{T\alpha_p\gamma} + 64d^{\frac{1}{p}-\frac{1}{2}}\frac{L_0\gamma}{\alpha_p^2}\log(1/\delta) + \frac{32\sigma}{\alpha_p B^{\frac{\kappa-1}{\kappa}}} + \frac{48d^{\frac{1}{p}-\frac{1}{2}}\|\nabla f(x^1)\|_2}{T\alpha_p^2}\log(\frac{1}{\delta}). \tag{22}$$

The $T$ dependence in the first three terms is dominant in comparison with the last term, hence, we neglect it. Next, we pick optimal parameters for optimal convergence.

**Optimal tuning, $\varepsilon \geq \frac{4L_0}{L_1}$:** Choosing the largest possible $\gamma = \frac{\alpha_p^2}{48d^{\frac{1}{p}-\frac{1}{2}}L_1^\delta}$, we bound the term:

$$64d^{\frac{1}{p}-\frac{1}{2}}\frac{L_0^\delta\gamma}{\alpha_p^2} \leq \frac{64}{48}\frac{L_0}{L_1} \leq \frac{\varepsilon}{2}.$$

The bound (22) becomes:

$$\frac{1}{T}\sum_{k=1}^{T}\|\nabla f(x^k)\|_p \leq \frac{512L_1^\delta d^{\frac{1}{p}-\frac{1}{2}}}{\alpha_p^3}\frac{\Delta}{T} + \frac{4\sigma}{\alpha_p B^{\frac{\kappa-1}{\kappa}}} + \frac{\varepsilon}{2}.$$

Choosing $B$ such that $\frac{4\sigma}{\alpha_p B^{\frac{\kappa-1}{\kappa}}} \leq \varepsilon/4 \Rightarrow B = \max\left\{1, \left(\frac{16\sigma}{\alpha_p\varepsilon}\right)^{\frac{\kappa}{\kappa-1}}\right\}$, we only need to bound

$$\frac{512L_1^\delta d^{\frac{1}{p}-\frac{1}{2}}}{\alpha_p^3}\frac{\Delta}{T} \leq \frac{\varepsilon}{4} \Rightarrow T = O\left(\frac{\Delta L_1^\delta d^{\frac{1}{p}-\frac{1}{2}}}{\alpha_p^3\varepsilon}\right).$$

The final sample complexity is

$$N = T\cdot B = O\left(\frac{\Delta L_1^\delta d^{\frac{1}{p}-\frac{1}{2}}}{\alpha_p^3\varepsilon}\left[1 + \left(\frac{\sigma}{\alpha_p\varepsilon}\right)^{\frac{\kappa}{\kappa-1}}\right]\right).$$

**Optimal tuning, $\varepsilon \leq \frac{4L_0}{L_1}$:** Choosing $B$ such that $\frac{4\sigma}{\alpha_p B^{\frac{\kappa-1}{\kappa}}} \leq \varepsilon/4 \Rightarrow B = \max\left\{1, \left(\frac{16\sigma}{\alpha_p\varepsilon}\right)^{\frac{\kappa}{\kappa-1}}\right\}$, we transform the bound (22) into

$$\frac{1}{T}\sum_{k=1}^{T}\|\nabla f(x^k)\|_p \leq \frac{2\Delta}{T\alpha_p\gamma} + 64d^{\frac{1}{p}-\frac{1}{2}}\frac{L_0^\delta\gamma}{\alpha_p^2} \leq \varepsilon.$$

Using $\gamma = \sqrt{\frac{\Delta}{32TL_0^\delta d^{\frac{1}{p}-\frac{1}{2}}}}$, we obtain

$$\frac{2\Delta}{T\alpha_p\gamma} + 64d^{\frac{1}{p}-\frac{1}{2}}\frac{L_0^\delta\gamma}{\alpha_p^2} = 16\sqrt{\frac{\Delta L_0^\delta d^{\frac{1}{p}-\frac{1}{2}}}{\alpha_p^3 T}} \leq \varepsilon.$$

Hence, required number of iterations $T = O\left(\frac{\Delta L_0^\delta d^{\frac{1}{p}-\frac{1}{2}}}{\alpha_p^3\varepsilon^2}\right)$ and the final comparison complexity is

$$N = T\cdot B = O\left(\frac{\Delta L_0^\delta d^{\frac{1}{p}-\frac{1}{2}}}{\alpha_p^3\varepsilon^2}\left[1 + \left(\frac{\sigma}{\alpha_p\varepsilon}\right)^{\frac{\kappa}{\kappa-1}}\right]\right).$$

We also notice that $\gamma = \sqrt{\frac{\Delta}{32TL_0^\delta d^{\frac{1}{p}-\frac{1}{2}}}} \leq \frac{\alpha_p^2}{48d^{\frac{1}{p}-\frac{1}{2}}L_1^\delta}$ for this number of iterations and $\varepsilon \leq \frac{4L_0}{L_1}$. $\quad\square$

**Theorem 3** (**HP complexity for** minibatch-CompSGD, **infinite horizon, Lipschitz noise**). *Consider lower-bounded $(L_0, L_1)$-smooth function $f$ (As. 1, 2) and HT Lipschitz function estimates $\kappa \in (1,2]$ (As. 3). Then Alg. 3 requires the sample complexity $N$ to achieve $\min\limits_{k \in \overline{1,T}} \|\nabla f(x^k)\|_p \leq \varepsilon$ with probability at least $1 - \delta$:*

*Parameter-free tuning: Until plateau $B_k = B_0 k^2, \gamma_k = \gamma_0 \leq \frac{\alpha_p^2}{48 L_1^{\delta,p}}$ and after $B_k = B_0 k, \gamma_k = \gamma_0/\sqrt{k}$ :*

$$
\varepsilon \geq \frac{4L_0}{L_1} \quad \Rightarrow \quad N = \tilde{O}\left( B_0 \left( \frac{\Delta}{\gamma_0 \alpha_p \varepsilon} \right)^3 + \frac{1}{B_0^2} \left( \frac{\sigma}{\alpha_p \varepsilon} \right)^{\frac{3\kappa}{2(\kappa-1)}} \right),
$$

$$
\varepsilon \ll \frac{4L_0}{L_1} \quad \Rightarrow \quad N = \tilde{O}\left( \frac{B_0 (L_0^{\delta,p} \gamma_0 + \Delta/\gamma_0)^4}{\alpha_p^8 \varepsilon^4} + \frac{1}{B_0} \left( \frac{\sigma}{\alpha_p \varepsilon} \right)^{\frac{2\kappa}{\kappa-1}} \right),
$$

*where $\Delta = f(x^1) - f^*, L_0^{\delta,p} = L_0 d^{\frac{1}{p} - \frac{1}{2}} \log(1/\delta), L_1^{\delta,p} = L_1 d^{\frac{1}{p} - \frac{1}{2}} \log(1/\delta).$*

*Proof.* First, we derive upper bound for new $\min$ metric with non-constant parameters from the bound (6) from Convergence Lemma 1:

$$
\min_{k \in \overline{1,T}} \|\nabla f(x^k)\|_p \leq \frac{\sum_{k=1}^T \gamma_k \|\nabla f(x^k)\|_p}{\sum_{k=1}^T \gamma_k} = \frac{8\Delta}{\alpha_p \sum_{k=1}^T \gamma_k} + 64 L_0 \frac{\sum_{k=1}^T \gamma_k^2}{\alpha_p \sum_{k=1}^T \gamma_k} + \frac{32 \sum_{k=1}^T \sigma \gamma_k / B_k^{\frac{\kappa-1}{\kappa}}}{\alpha_p \sum_{k=1}^T \gamma_k}
$$

$$
+ \frac{48 d^{\frac{1}{p} - \frac{1}{2}}}{\alpha_p} (\gamma_1 \|\nabla f(x^1)\|_1 + 2 C_T L_0) \frac{\log(1/\delta)}{\alpha_p \sum_{k=1}^T \gamma_k}.
$$

**Parameter-free tuning,** $\varepsilon \geq \frac{4L_0}{L_1}$**:** If we consider only first $T$ steps until plateau $\frac{4L_0}{L_1}$, we use constant stepsizes $\gamma_k = \gamma_0 \leq \frac{\alpha_p^2}{48 L_1^{\delta,p}}$ and increasing batchsizes $B_k = B_0 k^2$ to get

$$
\sum_{k=1}^T \gamma_k = T\gamma_0, \sum_{k=1}^T \gamma_k^2 = T\gamma_0^2, \gamma_1 = \gamma_0, C_T = T\gamma_0,
$$

$$
\sum_{k=1}^T \frac{\gamma_0}{B_k^{\frac{\kappa-1}{\kappa}}} = \sum_{k=1}^T \frac{\gamma_0}{(\sqrt{B_0}k)^{\frac{2(\kappa-1)}{\kappa}}} \leq \frac{\gamma_0 T^{\frac{2-\kappa}{\kappa}} \ln T}{B_0^{\frac{\kappa-1}{\kappa}}},
$$

$$
\min_{k \in \overline{1,T}} \|\nabla f(x^k)\|_p \leq \frac{8\Delta}{\alpha_p \gamma_0 T} + \frac{64 L_0^{\delta,p} \gamma_0}{\alpha_p^2} + \frac{32\sigma}{\alpha_p (T\sqrt{B_0})^{\frac{2(\kappa-1)}{\kappa}}} \ln T \leq \varepsilon.
$$

The term $\frac{64 L_0^{\delta,p} \gamma_0}{\alpha_p^2} \leq \frac{\varepsilon}{16}$ is bounded by condition, and the number of iterations $T = \tilde{O}\left( \left( \frac{\Delta}{\gamma_0 \alpha_p \varepsilon} \right) + \frac{1}{\sqrt{B_0}} \left( \frac{\sigma}{\alpha_p \varepsilon} \right)^{\frac{\kappa}{2(\kappa-1)}} \right)$ is enough to bound the other terms. The total sample complexity is

$$
\sum_{k=1}^T B_k = \sum_{k=1}^T B_0 k^2 \leq B_0 T^3 = \tilde{O}\left( B_0 \left( \frac{\Delta}{\gamma_0 \alpha_p \varepsilon} \right)^3 + \frac{1}{\sqrt{B_0}} \left( \frac{\sigma}{\alpha_p \varepsilon} \right)^{\frac{3\kappa}{2(\kappa-1)}} \right).
$$

**Case** $\varepsilon \ll \frac{4L_0}{L_1}$, **parameter-free tuning.** In this case, the first steps can be neglected, as we use decreasing stepsizes $\gamma_k = \frac{\gamma_0}{\sqrt{k}}$ and increasing batchsizes $B_k = B_0 k$ to get

$$\sum_{k=1}^{T} \gamma_k = \gamma_0 \sqrt{T}, \sum_{k=1}^{T} \gamma_k^2 = \gamma_0^2 \ln T, \gamma_1 = \gamma_0, C_T = \gamma_0^2,$$

$$\sum_{k=1}^{T} \frac{\gamma_k}{B_k^{\frac{\kappa-1}{\kappa}}} = \frac{\gamma_0}{B_0^{\frac{\kappa-1}{\kappa}}} \sum_{k=1}^{T} \frac{1}{k^{\frac{3\kappa-2}{2\kappa}}} \leq \frac{\gamma_0}{B_0^{\frac{\kappa-1}{\kappa}}} T^{\frac{2-\kappa}{2\kappa}} \ln T,$$

$$\min_{k \in \overline{1,T}} \|\nabla f(x^k)\|_p \leq \frac{8\Delta}{\alpha_p \gamma_0 \sqrt{T}} + 64 L_0^{\delta,p} \gamma_0 \frac{\ln T}{\alpha_p^2 \sqrt{T}} + \frac{32\sigma \ln T}{\alpha_p B_0^{\frac{\kappa-1}{\kappa}} T^{\frac{\kappa-1}{\kappa}}} \leq \varepsilon.$$

Hence, the number of iterations $T = \tilde{O}\left(\frac{(L_0^{\delta,p}\gamma_0 + \Delta/\gamma_0)^2}{\alpha_p^4 \varepsilon^2} + \frac{1}{B_0}\left(\frac{\sigma}{\alpha_p \varepsilon}\right)^{\frac{\kappa}{\kappa-1}}\right)$ is enough to bound the sum. The total sample complexity is

$$\sum_{k=1}^{T} B_k = \sum_{k=1}^{T} B_0 k \leq B_0 T^2 = \tilde{O}\left(\frac{B_0(L_0^{\delta,p}\gamma_0 + \Delta/\gamma_0)^4}{\alpha_p^8 \varepsilon^4} + \frac{1}{B_0}\left(\frac{\sigma}{\alpha_p \varepsilon}\right)^{\frac{2\kappa}{\kappa-1}}\right). \tag{23}$$

$\square$

### A.6 PROOF OF MajorityVote-CompSGD COMPLEXITY THEOREM 1, INDEPENDENT NOISE

We start the proof with the general lemma on convergence of MajorityVote-CompSGD. The proof of Theorem 1 in case of independent noise is located after the lemma and divided into two parts: for finite horizon with optimal tuning and for infinite horizon with parameter-free tuning (Theorem 4).

**Lemma 5** (MajorityVote-CompSGD **Convergence Lemma**). *Consider lower-bounded $(L_0, L_1)$-smooth function $f$ (As. 1, 2), random directions (As. 4) and function estimates with **HT, unimodal and symmetric** noise $\kappa > 0$ (As. 3). Then Alg. 2 after $T$ iterations with non-increasing stepsizes $\gamma_k \leq \alpha_p^2/(48L_1 d^{\frac{1}{p}-\frac{1}{2}} \log\frac{1}{\delta})$ and batchsize $M_k \geq 160/\kappa^2$ achieves with probability at least $1 - \delta$ starting with $\Delta := f(x^1) - f^*$:*

$$\frac{\alpha_p}{8} \sum_{k=1}^{T} \gamma_k \|\nabla f(x^k)\|_p \leq 8\Delta + 64L_0 \sum_{k=1}^{T} \gamma_k^2 + 32 \sum_{k=1}^{T} \frac{\tilde{\sigma}_k}{\sqrt{M_k}} + 48 \frac{d^{\frac{1}{p}-\frac{1}{2}}}{\alpha_p}(\gamma_1 \|\nabla f(x^1)\|_p + C_T L_0) \log(\frac{1}{\delta}),$$
$$\tag{24}$$

*where $\tilde{\sigma}_k = \sigma$ for independent noise and $\tilde{\sigma}_k = \gamma_k \sigma$ for Lipschitz noise, $C_T := \max_{k \in \overline{1,T}} \gamma_k \cdot \sum_{\tau=1}^{k-1} \gamma_\tau$.*

*Proof.* The beginning of the proof copies the proof of CompSGD Convergence Lemma 1 from Appendix A.3 until the line (38) where we now need to estimate probability

$$\mathbb{P}_\xi \left[ \text{sign} \left[ \sum_{i=1}^{M} \text{sign}(f(x^k + \gamma_k \mathbf{e}^k, \xi_{i,+}^k) - f(x^k - \gamma_k \mathbf{e}^k, \xi_{i,-}^k)) \right] \neq \text{sign}(\langle \nabla f(x^k), \mathbf{e}^k \rangle) \right] = (*).$$

Each comparison $\text{sign}(f(x^k + \gamma_k \mathbf{e}^k, \xi_{i,+}^k) - f(x^k - \gamma_k \mathbf{e}^k, \xi_{i,-}^k)) \neq \text{sign}(\langle \nabla f(x^k), \mathbf{e}^k \rangle)$ is a Bernoulli trial with failure probability (40):

$$\mathbb{P}_\xi \left[\text{sign}(f(x^k + \gamma_k \mathbf{e}^k, \xi_{i,+}^k) - f(x^k - \gamma_k \mathbf{e}^k, \xi_{i,-}^k)) \neq \text{sign}(\langle \nabla f(x^k), \mathbf{e}^k \rangle) \right] \leq \mathbb{P}_\xi \left[|\theta_{i,+}^k - \theta_{i,-}^k| \geq \gamma_k \cdot |\langle \nabla f(x^k), \mathbf{e}^k \rangle| \right].$$

The rhs probability can be estimated using Gauss inequality for unimodal symmetric noise $\theta_{i,+}^k - \theta_{i,-}^k$ by the generalized Gauss's Inequality(Dharmadhikari & Joag-Dev, 1986, Theorem 1).

**Lemma 6** (Gauss's Inequality). *Let a random variable $\xi$ be unimodal symmetric with mode $\nu$ and bounded $\kappa$-th moment, $\kappa > 0$. Then the following bounds hold:*

$$\mathbb{P}[|\xi - \nu| \geq \tau] \leq \begin{cases} \left(\frac{\kappa}{\kappa+1}\right)^\kappa \frac{\mathbb{E}[|\xi-\nu|]^\kappa}{\tau^\kappa}, & \tau^\kappa \geq \frac{\kappa^\kappa}{(\kappa+1)^{\kappa-1}} \cdot \mathbb{E}[|\xi - \nu|^\kappa], \\ 1 - \left[\frac{\tau^\kappa}{(\kappa+1)\mathbb{E}[|\xi-\nu|]^\kappa}\right]^{\frac{1}{\kappa}}, & \tau^\kappa \leq \frac{\kappa^\kappa}{(\kappa+1)^{\kappa-1}} \cdot \mathbb{E}[|\xi - \nu|^\kappa]. \end{cases}$$

We denote $S = \frac{\gamma_k |\langle \nabla f(x^k), \mathbf{e}^k \rangle|}{2 \tilde{\sigma}_k}$, $q := \mathbb{P}_\xi \left[ |\theta_{i,+}^k - \theta_{i,-}^k| \geq \gamma_k \cdot |\langle \nabla f(x^k), \mathbf{e}^k \rangle| \right]$ and estimate

$$q = \mathbb{P}_\xi \left[ |\theta_{i,+}^k - \theta_{i,-}^k| \geq \gamma_k \cdot |\langle \nabla f(x^k), \mathbf{e}^k \rangle| \right] \leq \begin{cases} \frac{1}{2} \left( \frac{\kappa}{\kappa+1} \right)^\kappa \frac{1}{S^\kappa}, & S^\kappa \geq \frac{\kappa^\kappa}{(\kappa+1)^{\kappa-1}}, \\ \frac{1}{2} - \frac{1}{2} \frac{S}{(\kappa+1)^{\frac{1}{\kappa}}}, & S^\kappa \leq \frac{\kappa^\kappa}{(\kappa+1)^{\kappa-1}}. \end{cases}$$

We denote probability of failure of a single estimate by

$$q \leq \begin{cases} \frac{1}{2} \left( \frac{\kappa}{\kappa+1} \right)^\kappa \frac{1}{S^\kappa}, & S^\kappa \geq \frac{\kappa^\kappa}{(\kappa+1)^{\kappa-1}}, \\ \frac{1}{2} - \frac{1}{2} \frac{S}{(\kappa+1)^{\frac{1}{\kappa}}}, & S^\kappa \leq \frac{\kappa^\kappa}{(\kappa+1)^{\kappa-1}}, \end{cases} =: \tilde{q}(S).$$

Moreover, probability $q \leq \tilde{q}(S) < \frac{1}{2}$, and the deviation of $q$ from $\frac{1}{2}$ can be bounded by

$$\varepsilon := \frac{1}{2} - q \leq \frac{1}{2} - \tilde{q}(S) =: \tilde{\varepsilon}(S).$$

The probability of getting the wrong sign can be restated as the probability of failing half out of $M_k$ Bernoulli trials with fail probability $q_j$:

$$(*) \leq \frac{1}{1 + \frac{M_k}{\frac{1}{4\varepsilon^2} - 1}} < \frac{1}{1 + \frac{M_k}{\frac{1}{4\tilde{\varepsilon}^2(S)} - 1}}. \tag{25}$$

- First, we consider the case $S \geq \frac{\kappa}{(\kappa+1)^{\frac{\kappa-1}{\kappa}}}$:

$$\tilde{\varepsilon}^2(S) = \left( \frac{1}{2} - \frac{1}{2} \left( \frac{\kappa}{\kappa+1} \right)^\kappa \frac{1}{S^\kappa} \right)^2 \geq \frac{1}{4} \frac{\kappa^2}{(\kappa+1)^2},$$

$$\frac{1}{4\tilde{\varepsilon}^2(S)} - 1 \leq \frac{(\kappa+1)^2}{\kappa^2} - 1 \leq \frac{5}{\kappa^2}.$$

If we set $M_k \geq \frac{160}{\kappa^2}$, then the fail probability is upper bounded by

$$(*) < \frac{1}{1 + \frac{M_k}{\frac{1}{4\tilde{\varepsilon}^2(S)} - 1}} \leq \frac{1}{32}. \tag{26}$$

- For the case $S < \frac{\kappa}{(\kappa+1)^{\frac{\kappa-1}{\kappa}}}$, we derive the bound:

$$\frac{1}{4\tilde{\varepsilon}^2(S)} - 1 = \frac{(\kappa+1)^{\frac{2}{\kappa}}}{S^2} - 1 \leq \frac{4}{S^2}. \tag{27}$$

And we use the inequality $\frac{1}{1+x^2} \leq \frac{1}{2x}, x > 0$ on (25):

$$(25) \leq \frac{\sqrt{\frac{1}{4\tilde{\varepsilon}^2(S)} - 1}}{2\sqrt{M_k}} \leq \frac{1}{\sqrt{M_k}} \cdot \frac{1}{S}. \tag{28}$$

Combining (26) and (28) together, we obtain the bound for each coordinate:

$$\mathbb{P}_\xi \left[ \text{sign}(f(x^k + \gamma_k \mathbf{e}^k, \xi_{i,+}^k) - f(x^k - \gamma_k \mathbf{e}^k, \xi_{i,-}^k)) \neq \text{sign}(\langle \nabla f(x^k), \mathbf{e}^k \rangle) \right]$$
$$\leq \frac{1}{32} + \frac{1}{\sqrt{M_k}} \cdot \frac{1}{S_j} = \frac{1}{32} + \frac{1}{\sqrt{M_k}} \frac{2\tilde{\sigma}_k}{\gamma_k |\langle \nabla f(x^k), \mathbf{e}^k \rangle|}. \tag{29}$$

The rest of the proof copies the proof of CompSGD Convergence Lemma 1 from Appendix A.3 with substitution $\tilde{\sigma}_k \to \frac{\tilde{\sigma}_k}{\sqrt{M_k}}$, and in the end we obtain the bound:

$$\frac{\alpha_p}{8} \sum_{k=1}^T \gamma_k \|\nabla f(x^k)\|_p \leq 8\Delta + 64 L_0 \sum_{k=1}^T \gamma_k^2 + 32 \sum_{k=1}^T \frac{\tilde{\sigma}_k}{\sqrt{M_k}} + 48 \frac{d^{\frac{1}{p} - \frac{1}{2}}}{\alpha_p} (\gamma_1 \|\nabla f(x^1)\|_p + C_T L_0) \log(\frac{1}{\delta}).$$

$\square$

*Proof of* MajorityVote-CompSGD *Complexity Theorem 1, independent noise.* Plugging in constant stepsizes and batchsizes $\gamma_k \equiv \gamma$, $C_T = T\gamma^2$, $\gamma_1 = \gamma$ in the convergence bound (24) from Convergence Lemma 5 and dividing both sides by $\frac{\alpha_p T \gamma}{8}$, we obtain the bound for independent noise:

$$\frac{1}{T} \sum_{k=1}^{T} \|\nabla f(x^k)\|_p \leq \frac{8\Delta}{T\alpha_p\gamma} + 64 d^{\frac{1}{p}-\frac{1}{2}} \frac{L_0\gamma}{\alpha_p^2} \log(1/\delta) + \frac{32\sigma}{\alpha_p\gamma\sqrt{M}} + \frac{48 d^{\frac{1}{p}-\frac{1}{2}} \|\nabla f(x^1)\|_2}{T\alpha_p^2} \log(\frac{1}{\delta}). \tag{30}$$

**Optimal tuning, $\varepsilon \geq \frac{4L_0}{L_1}$:** Choosing the largest possible $\gamma = \frac{\alpha_p^2}{48 d^{\frac{1}{p}-\frac{1}{2}} L_1^\delta}$, we bound the term:

$$64 d^{\frac{1}{p}-\frac{1}{2}} \frac{L_0^\delta\gamma}{\alpha_p^2} \leq \frac{64}{48} \frac{L_0}{L_1} \leq \frac{3\varepsilon}{2}.$$

The bound (30) becomes:

$$\frac{1}{T} \sum_{k=1}^{T} \|\nabla f(x^k)\|_p \leq \frac{512 L_1^\delta d^{\frac{1}{p}-\frac{1}{2}}}{\alpha_p^3} \left[ \frac{\Delta}{T} + \frac{4\sigma}{\sqrt{M}} \right] + \frac{3\varepsilon}{2}.$$

Choosing $M$ such that $\frac{4\sigma}{\Delta\sqrt{M}} \leq \frac{1}{T} \Rightarrow M = \max\left\{ \frac{160}{\kappa^2}, \left(\frac{4\sigma T}{\Delta}\right)^2 \right\}$, we only need to bound

$$\frac{1024 L_1^\delta d^{\frac{1}{p}-\frac{1}{2}}}{\alpha_p^3} \frac{\Delta}{T} \leq \frac{\varepsilon}{2} \Rightarrow T = O\left( \frac{\Delta L_1^\delta d^{\frac{1}{p}-\frac{1}{2}}}{\alpha_p^3\varepsilon} \right).$$

The final sample complexity is

$$N = T \cdot M = \frac{\Delta L_1^\delta d^{\frac{1}{p}-\frac{1}{2}}}{\alpha_p^3\varepsilon} \left[ \frac{1}{\kappa^2} + \left( \frac{\sigma L_1^\delta d^{\frac{1}{p}-\frac{1}{2}}}{\alpha_p^3\varepsilon} \right)^2 \right].$$

**Optimal tuning, $\varepsilon \leq \frac{4L_0}{L_1}$:** Choosing $M$ such that $\frac{4\sigma}{\Delta\sqrt{M}} \leq \frac{1}{T} \Rightarrow M = \max\left\{ \frac{160}{\kappa^2}, \left(\frac{4\sigma T}{\Delta}\right)^2 \right\}$, we transform the bound (30) into

$$\frac{1}{T} \sum_{k=1}^{T} \|\nabla f(x^k)\|_p \leq \frac{2\Delta}{T\alpha_p\gamma} + 64 d^{\frac{1}{p}-\frac{1}{2}} \frac{L_0^\delta\gamma}{\alpha_p^2} \leq \varepsilon.$$

Using $\gamma = \sqrt{\frac{\alpha_p\Delta}{32 T L_0^\delta d^{\frac{1}{p}-\frac{1}{2}}}}$, we obtain

$$\frac{2\Delta}{T\alpha_p\gamma} + 64 d^{\frac{1}{p}-\frac{1}{2}} \frac{L_0^\delta\gamma}{\alpha_p^2} = 16\sqrt{\frac{\Delta L_0^\delta d^{\frac{1}{p}-\frac{1}{2}}}{\alpha_p^3 T}} \leq \varepsilon.$$

Hence, required number of iterations $T = O\left( \frac{\Delta L_0^\delta d^{\frac{1}{p}-\frac{1}{2}}}{\alpha_p^3\varepsilon^2} \right)$ and the final comparison complexity is

$$N = T \cdot M = \frac{\Delta L_0^\delta d^{\frac{1}{p}-\frac{1}{2}}}{\alpha_p^3\varepsilon^2} \left[ \frac{1}{\kappa^2} + \left( \frac{\sigma L_0^\delta d^{\frac{1}{p}-\frac{1}{2}}}{\alpha_p^3\varepsilon^2} \right)^2 \right].$$

We also notice that $\gamma = \sqrt{\frac{\alpha_p\Delta}{32 T L_0^\delta d^{\frac{1}{p}-\frac{1}{2}}}} \leq \frac{\alpha_p^2}{48 d^{\frac{1}{p}-\frac{1}{2}} L_1^\delta}$ for this number of iterations and $\varepsilon \leq \frac{4L_0}{L_1}$. $\square$

**Theorem 4 (HP complexity for MajorityVote-CompSGD, infinite horizon, independent noise).** *Consider lower-bounded $(L_0, L_1)$-smooth function $f$ (As. 1, 2) and HT function estimates corrupted by **independent, unimodal and symmetric HT noise** with $\kappa > 0$ (As. 3). Then Alg. 2 requires the sample complexity $N$ to achieve $\min_{k \in \overline{1,T}} \|\nabla f(x^k)\|_p \leq \varepsilon$ with probability at least $1 - \delta$ for:*

**Parameter-free tuning:** $M_k = M_0(k/\kappa)^2$, $\gamma_k = \gamma_0 \leq \frac{\alpha_p^2}{48L_1^{\delta,p}}$ until plateau $\gamma_k = \frac{\gamma_0}{\sqrt{k}}$ after:

$$\varepsilon \geq \frac{4L_0}{L_1} \quad \Rightarrow \quad N = \tilde{O}\left(\frac{M_0(L_1^{\delta,p}(\Delta + \sigma/\sqrt{M_0}))^3}{\alpha_p^3 \kappa^2 \varepsilon^3}\right),$$

$$\varepsilon \ll \frac{4L_0}{L_1} \quad \Rightarrow \quad N = \tilde{O}\left(\frac{M_0}{\kappa^2}\left(\frac{(\Delta + \sigma/\sqrt{M_0})/\gamma_0 + L_0^{\delta,p}\gamma_0}{\alpha_p^2 \varepsilon}\right)^6\right).$$

where $\Delta = f(x^1) - f^*$, $L_0^{\delta,p} = L_0 d^{\frac{1}{p}-\frac{1}{2}}\log(1/\delta)$, $L_1^{\delta,p} = L_1 d^{\frac{1}{p}-\frac{1}{2}}\log(1/\delta)$.

*Proof.* First, we derive upper bound for new $\min$ metric with non-constant parameters from (24):

$$\min_{k \in \overline{1,T}} \|\nabla f(x^k)\|_p \leq \frac{\sum_{k=1}^{T} \gamma_k \|\nabla f(x^k)\|_p}{\sum_{k=1}^{T} \gamma_k} = \frac{8\Delta}{\alpha_p \sum_{k=1}^{T} \gamma_k} + 64L_0 \frac{\sum_{k=1}^{T} \gamma_k^2}{\alpha_p \sum_{k=1}^{T} \gamma_k} + \frac{32\sum_{k=1}^{T} \sigma/\sqrt{M_k}}{\alpha_p \sum_{k=1}^{T} \gamma_k}$$

$$+ \quad 6d^{\frac{1}{p}-\frac{1}{2}}(\gamma_1 \|\nabla f(x^1)\|_1 + 2C_T L_0)\frac{\log(1/\delta)}{\alpha_p^2 \sum_{k=1}^{T} \gamma_k}.$$

**Parameter-free tuning, $\varepsilon \geq \frac{4L_0}{L_1}$:** If we consider only first $T$ steps until plateau $\frac{4L_0}{L_1}$, we use constant stepsizes $\gamma_k = \gamma_0 \leq \frac{\alpha_p^2}{48L_1^{\delta,p}}$ and increasing batchsizes $M_k = M_0(k/\kappa)^2$ to get

$$\sum_{k=1}^{T} \gamma_k = T\gamma_0, \sum_{k=1}^{T} \gamma_k^2 = T\gamma_0^2, \gamma_1 = \gamma_0, C_T = T\gamma_0,$$

$$\sum_{k=1}^{T} \frac{1}{\sqrt{M_k}} = \sum_{k=1}^{T} \frac{\kappa}{\sqrt{M_0}k} \leq \frac{\kappa}{\sqrt{M_0}}\ln T,$$

$$\min_{k \in \overline{1,T}} \|\nabla f(x^k)\|_1 \leq \frac{8\Delta}{\alpha_p \gamma_0 T} + \frac{64L_0^{\delta,p}\gamma_0}{\alpha_p^2} + \frac{32\sigma}{\alpha_p \gamma_0 \sqrt{M_0}T}\kappa \ln T \leq \varepsilon.$$

The term $\frac{64L_0^{\delta,p}\gamma_0}{\alpha_p^2} \leq \frac{\varepsilon}{4}$ is bounded by condition, and the number of iterations $T = \tilde{O}\left(\frac{(\sigma/\sqrt{M_0}+\Delta)}{\alpha_p \gamma_0 \varepsilon}\right)$ is enough to bound the other terms. The total sample complexity is

$$\sum_{k=1}^{T} M_k = \sum_{k=1}^{T} M_0(k/\kappa)^2 \leq M_0 T^3/\kappa^2 = \tilde{O}\left(\frac{M_0}{\kappa^2}\left(\frac{\sigma/\sqrt{M_0}+\Delta}{\alpha_p \gamma_0 \varepsilon}\right)^3\right).$$

**Parameter-free tuning, $\varepsilon \ll \frac{4L_0}{L_1}$:** In this case, the first steps can be neglected, since we use decreasing stepsizes $\gamma_k = \frac{\gamma_0}{\sqrt{k}}$ and increasing batchsizes $M_k = M_0(k/\kappa)^2$ to get

$$\sum_{k=1}^{T} \gamma_k = \gamma_0\sqrt{T}, \sum_{k=1}^{T} \gamma_k^2 = \gamma_0^2 \ln T, \gamma_1 = \gamma_0, C_T = \gamma_0^2,$$

$$\sum_{k=1}^{T} \frac{1}{\sqrt{M_k}} = \sum_{k=1}^{T} \frac{\kappa}{\sqrt{M_0}k} \leq \frac{\kappa}{\sqrt{M_0}}\ln T,$$

$$\min_{k \in \overline{1,T}} \|\nabla f(x^k)\|_p \leq \frac{8\Delta}{\alpha_p \gamma_0 \sqrt{T}} + 64L_0^{\delta,p}\gamma_0 \frac{\ln T}{\alpha_p^2 \sqrt{T}} + \frac{32\kappa\sigma \ln T}{\alpha_p \gamma_0 \sqrt{M_0}T} \leq \varepsilon.$$

Hence, the number of iterations $T = \tilde{O}\left(\left(\frac{(\Delta+\sigma/\sqrt{M_0})/\gamma_0 + L_0^{\delta,p}\gamma_0}{\alpha_p^2 \varepsilon}\right)^2\right)$ is enough to bound the sum. The total sample complexity is

$$\sum_{k=1}^{T} M_k = \sum_{k=1}^{T} M_0(k/\kappa)^2 \leq M_0 T^3/\kappa^2 = \tilde{O}\left(\frac{M_0}{\kappa^2}\left(\frac{(\Delta + \sigma/\sqrt{M_0})/\gamma_0 + L_0^{\delta,p}\gamma_0}{\alpha_p^2 \varepsilon}\right)^6\right). \quad (31)$$

$\square$

### A.7 PROOF OF MajorityVote-CompSGD COMPLEXITY THEOREM 1, LIPSCHITZ NOISE

The proof of Theorem 1 in case of Lipschitz noise is divided into two parts: for finite horizon with optimal tuning (Theorem 5) and for infinite horizon with parameter-free tuning (Theorem 6).

**Theorem 5** (**HP complexity for** MajorityVote-CompSGD, **finite horizon, Lipschitz noise**). *Consider the lower-bounded $(L_0, L_1)$-smooth function $f$ (As. 1, 2), random directions with $\alpha_p$ (As. 4) and function estimates with HT **Lipschitz, unimodal and symmetric** noise $\kappa > 0$ (As. 3). Then Alg. 2 requires comparison number $N$ to achieve $\frac{1}{T}\sum_{k=1}^{T}\|\nabla f(x_k)\|_p \leq \varepsilon$ with probability at least $1 - \delta$ for:*

***Optimal tuning:*** $T = O\left(\frac{\Delta L_1^{\delta,p}}{\alpha_p^3\varepsilon}\right), \gamma_k = \frac{\alpha_p^2}{48L_1^{\delta,p}}, M_k = \max\left\{\frac{160}{\kappa^2}, \left(\frac{128\sigma}{\alpha_p\varepsilon}\right)^2\right\}$ *for* $\varepsilon \geq \frac{4L_0}{L_1}$ *and*

$T = O\left(\frac{\Delta L_0^{\delta,p}}{\alpha_p^3\varepsilon^2}\right), \gamma_k \equiv \sqrt{\frac{\alpha_p\Delta}{32TL_0^{\delta,p}}}, M_k = \max\left\{\frac{160}{\kappa^2}, \left(\frac{128\sigma}{\alpha_p\varepsilon}\right)^2\right\}$ *for* $\varepsilon \leq \frac{4L_0}{L_1}$ :

$$N = O\left(\left(\frac{\Delta L_1^{\delta,p}}{\alpha_p^3\varepsilon} + \frac{\Delta L_0^{\delta,p}}{\alpha_p^3\varepsilon^2}\right)\left[\frac{1}{\kappa^2} + \frac{\sigma^2}{\alpha_p^2\varepsilon^2}\right]\right), \tag{32}$$

*where* $\Delta = f(x^1) - f^*, L_0^{\delta,p} = L_0 d^{\frac{1}{p}-\frac{1}{2}}\log(\frac{1}{\delta}), L_1^{\delta,p} = L_1 d^{\frac{1}{p}-\frac{1}{2}}\log(\frac{1}{\delta}).$

*Proof.* Plugging in constant stepsizes and batchsizes $\gamma_k \equiv \gamma, C_T = T\gamma^2, \gamma_1 = \gamma$ in the convergence bound (24) from Convergence Lemma 5 and dividing both sides by $\frac{\alpha_p T\gamma}{8}$, we obtain the bound for Lipschitz noise:

$$\frac{1}{T}\sum_{k=1}^{T}\|\nabla f(x^k)\|_p \leq \frac{8\Delta}{T\alpha_p\gamma} + 64d^{\frac{1}{p}-\frac{1}{2}}\frac{L_0\gamma}{\alpha_p^2}\log(1/\delta) + \frac{32\sigma}{\alpha_p\sqrt{M}} + \frac{48d^{\frac{1}{p}-\frac{1}{2}}\|\nabla f(x^1)\|_2}{T\alpha_p^2}\log(\frac{1}{\delta}). \tag{33}$$

The $T$ dependence in the first three terms is dominant in comparison with the last term, hence, we neglect it. Next, we pick optimal parameters for optimal convergence.

**Optimal tuning,** $\varepsilon \geq \frac{4L_0}{L_1}$**:** Choosing the largest possible $\gamma = \frac{\alpha_p^2}{48d^{\frac{1}{p}-\frac{1}{2}}L_1^{\delta}}$, we bound the term:

$$64d^{\frac{1}{p}-\frac{1}{2}}\frac{L_0^\delta\gamma}{\alpha_p^2} \leq \frac{64}{48}\frac{L_0}{L_1} \leq \frac{3\varepsilon}{2}.$$

The bound (33) becomes:

$$\frac{1}{T}\sum_{k=1}^{T}\|\nabla f(x^k)\|_p \leq \frac{512L_1^\delta d^{\frac{1}{p}-\frac{1}{2}}}{\alpha_p^3}\frac{\Delta}{T} + \frac{32\sigma}{\alpha_p\sqrt{M}} + \frac{\varepsilon}{2}.$$

Choosing $B$ such that $\frac{32\sigma}{\alpha_p\sqrt{M}} \leq \varepsilon/4 \Rightarrow M = \max\left\{160/\kappa^2, \left(\frac{128\sigma}{\alpha_p\varepsilon}\right)^2\right\}$, we only need to bound

$$\frac{512L_1^\delta d^{\frac{1}{p}-\frac{1}{2}}}{\alpha_p^3}\frac{\Delta}{T} \leq \frac{\varepsilon}{4} \Rightarrow T = O\left(\frac{\Delta L_1^\delta d^{\frac{1}{p}-\frac{1}{2}}}{\alpha_p^3\varepsilon}\right).$$

The final sample complexity is

$$N = T \cdot M = O\left(\frac{\Delta L_1^\delta d^{\frac{1}{p}-\frac{1}{2}}}{\alpha_p^3\varepsilon}\left[\frac{1}{\kappa^2} + \left(\frac{\sigma}{\alpha_p\varepsilon}\right)^2\right]\right).$$

**Optimal tuning,** $\varepsilon \leq \frac{4L_0}{L_1}$**:** Choosing $M$ such that $\frac{32\sigma}{\alpha_p\sqrt{M}} \leq \varepsilon/4 \Rightarrow M = \max\left\{160/\kappa^2, \left(\frac{128\sigma}{\alpha_p\varepsilon}\right)^2\right\}$, we transform the bound (33) into

$$\frac{1}{T}\sum_{k=1}^{T}\|\nabla f(x^k)\|_p \leq \frac{2\Delta}{T\alpha_p\gamma} + 64d^{\frac{1}{p}-\frac{1}{2}}\frac{L_0^\delta\gamma}{\alpha_p^2} \leq \varepsilon.$$

Using $\gamma = \sqrt{\frac{\alpha_p \Delta}{32 T L_0^\delta d^{\frac{1}{p}-\frac{1}{2}}}}$, we obtain

$$\frac{2\Delta}{T\alpha_p\gamma} + 64 d^{\frac{1}{p}-\frac{1}{2}} \frac{L_0^\delta \gamma}{\alpha_p^2} = 16\sqrt{\frac{\Delta L_0^\delta d^{\frac{1}{p}-\frac{1}{2}}}{\alpha_p^2 T}} \le \varepsilon.$$

Hence, required number of iterations $T = O\left(\frac{\Delta L_0^\delta d^{\frac{1}{p}-\frac{1}{2}}}{\alpha_p^3 \varepsilon^2}\right)$ and the final comparison complexity is

$$N = T \cdot M = O\left(\frac{\Delta L_0^\delta d^{\frac{1}{p}-\frac{1}{2}}}{\alpha_p^3 \varepsilon^2}\left[\frac{1}{\kappa^2} + \left(\frac{\sigma}{\alpha_p \varepsilon}\right)^2\right]\right).$$

We also notice that $\gamma = \sqrt{\frac{\alpha_p \Delta}{32 T L_0^\delta d^{\frac{1}{p}-\frac{1}{2}}}} \le \frac{\alpha_p^2}{48 d^{\frac{1}{p}-\frac{1}{2}} L_1^\delta}$ for this number of iterations and $\varepsilon \le \frac{4L_0}{L_1}$. $\square$

**Theorem 6** (**HP complexity for** MajorityVote-CompSGD, **infinite horizon, Lipschitz noise**). *Consider lower-bounded $(L_0, L_1)$-smooth function $f$ (As. 1, 2) and **HT, Lipschitz, unimodal, symmetric** function estimates $\kappa > 0$ (As. 3). Then Alg. 2 requires the sample complexity $N$ to achieve $\min_{k \in \overline{1,T}} \|\nabla f(x^k)\|_p \le \varepsilon$ with probability at least $1 - \delta$:*

*Parameter-free tuning: Until plateau $M_k = M_0 k^2/\kappa^2, \gamma_k = \gamma_0 \le \frac{\alpha_p^2}{48 L_1^{\delta,p}}$ and after $M_k = M_0 k/\kappa^2, \gamma_k = \gamma_0/\sqrt{k}$ :*

$$\varepsilon \ge \frac{4L_0}{L_1} \quad \Rightarrow \quad N = \tilde{O}\left(\frac{M_0}{\kappa^2}\left(\frac{\Delta/\gamma_0 + \sigma/\sqrt{M_0}}{\alpha_p \varepsilon}\right)^3\right),$$

$$\varepsilon \ll \frac{4L_0}{L_1} \quad \Rightarrow \quad N = \tilde{O}\left(\frac{M_0(L_0^{\delta,p}\gamma_0 + \Delta/\gamma_0)^4}{\kappa^2 \alpha_p^8 \varepsilon^4} + \frac{1}{\kappa^2 M_0}\left(\frac{\sigma}{\alpha_p \varepsilon}\right)^4\right),$$

*where $\Delta = f(x^1) - f^*, L_0^{\delta,p} = L_0 d^{\frac{1}{p}-\frac{1}{2}} \log(1/\delta), L_1^{\delta,p} = L_1 d^{\frac{1}{p}-\frac{1}{2}} \log(1/\delta)$.*

*Proof.* First, we derive upper bound for new min metric with non-constant parameters from (24):

$$
\begin{aligned}
\min_{k \in \overline{1,T}} \|\nabla f(x^k)\|_p &\le \frac{\sum_{k=1}^T \gamma_k \|\nabla f(x^k)\|_p}{\sum_{k=1}^T \gamma_k} = \frac{8\Delta}{\alpha_p \sum_{k=1}^T \gamma_k} + 64 L_0 \frac{\sum_{k=1}^T \gamma_k^2}{\alpha_p \sum_{k=1}^T \gamma_k} + \frac{32 \sum_{k=1}^T \sigma \gamma_k/\sqrt{M_k}}{\alpha_p \sum_{k=1}^T \gamma_k} \\
&+ 6 d^{\frac{1}{p}-\frac{1}{2}} (\gamma_1 \|\nabla f(x^1)\|_1 + 2 C_T L_0) \frac{\log(1/\delta)}{\alpha_p^2 \sum_{k=1}^T \gamma_k}.
\end{aligned}
$$

**Parameter-free tuning,** $\varepsilon \ge \frac{4L_0}{L_1}$: If we consider only first $T$ steps until plateau $\frac{4L_0}{L_1}$, we use constant stepsizes $\gamma_k = \gamma_0 \le \frac{\alpha_p^2}{48 L_1^{\delta,p}}$ and increasing batchsizes $M_k = M_0 k^2$ to get

$$\sum_{k=1}^T \gamma_k = T\gamma_0, \sum_{k=1}^T \gamma_k^2 = T\gamma_0^2, \gamma_1 = \gamma_0, C_T = T\gamma_0,$$

$$\sum_{k=1}^T \frac{\gamma_0}{\sqrt{M_k}} = \sum_{k=1}^T \frac{\kappa\gamma_0}{\sqrt{M_0}k} \le \frac{\kappa\gamma_0 \ln T}{\sqrt{M_0}},$$

$$\min_{k \in \overline{1,T}} \|\nabla f(x^k)\|_p \le \frac{8\Delta}{\alpha_p \gamma_0 T} + \frac{64 L_0^{\delta,p}\gamma_0}{\alpha_p^2} + \frac{32\sigma}{\alpha_p(T\sqrt{M_0})} \ln T \le \varepsilon.$$

The term $\frac{64L_0^{\delta,p}\gamma_0}{\alpha_p^2} \leq \frac{\varepsilon}{16}$ is bounded by condition, and the number of iterations $T = \tilde{O}\left(\frac{\Delta/\gamma_0 + \sigma/\sqrt{M_0}}{\alpha_p \varepsilon}\right)$ is enough to bound the other terms. The total sample complexity is

$$\sum_{k=1}^{T} M_k = \sum_{k=1}^{T} \frac{M_0}{\kappa^2} k^2 \leq \frac{M_0}{\kappa^2} T^3 = \tilde{O}\left(\frac{M_0}{\kappa^2}\left(\frac{\Delta/\gamma_0 + \sigma/\sqrt{M_0}}{\alpha_p \varepsilon}\right)^3\right).$$

**Case** $\varepsilon \ll \frac{4L_0}{L_1}$**, parameter-free tuning.** In this case, the first steps can be neglected, as we use decreasing stepsizes $\gamma_k = \frac{\gamma_0}{\sqrt{k}}$ and increasing batchsizes $M_k = M_0 k/\kappa^2$ to get

$$\sum_{k=1}^{T} \gamma_k = \gamma_0 \sqrt{T}, \sum_{k=1}^{T} \gamma_k^2 = \gamma_0^2 \ln T, \gamma_1 = \gamma_0, C_T = \gamma_0^2,$$

$$\sum_{k=1}^{T} \frac{\gamma_k}{\sqrt{M_k}} = \frac{\kappa \gamma_0}{\sqrt{M_0}} \sum_{k=1}^{T} \frac{1}{k} \leq \frac{\kappa \gamma_0}{\sqrt{M_0}} \ln T,$$

$$\min_{k \in \overline{1,T}} \|\nabla f(x^k)\|_p \leq \frac{8\Delta}{\alpha_p \gamma_0 \sqrt{T}} + 64L_0^{\delta,p} \gamma_0 \frac{\ln T}{\alpha_p^2 \sqrt{T}} + \frac{32\sigma \ln T}{\alpha_p \sqrt{M_0 T}} \leq \varepsilon.$$

Hence, the number of iterations $T = \tilde{O}\left(\frac{(L_0^{\delta,p}\gamma_0 + \Delta/\gamma_0)^2}{\alpha_p^4 \varepsilon^2} + \frac{1}{M_0}\left(\frac{\sigma}{\alpha_p \varepsilon}\right)^2\right)$ is enough to bound the sum. The total sample complexity is

$$\sum_{k=1}^{T} M_k = \sum_{k=1}^{T} \frac{M_0}{\kappa^2} k \leq \frac{M_0}{\kappa^2} T^2 = \tilde{O}\left(\frac{M_0(L_0^{\delta,p}\gamma_0 + \Delta/\gamma_0)^4}{\kappa^2 \alpha_p^8 \varepsilon^4} + \frac{1}{\kappa^2 M_0}\left(\frac{\sigma}{\alpha_p \varepsilon}\right)^4\right). \tag{34}$$

$\square$

## A.8 IN EXPECTATION CONVERGENCE OF minibatch-CompSGD

**Theorem 7 (In expectation complexity for minibatch-CompSGD).** *Consider lower-bounded $(L_0, L_1)$-smooth function $f$ (As. 1, 2), random directions (As. 4) and HT function estimates $\kappa \in (1, 2]$ (As. 3). Then Alg. 3 requires $N$ function calls to achieve $\frac{1}{T}\sum_{k=1}^{T} \mathbb{E}[\|\nabla f(x_k)\|_p] \leq \varepsilon$ with probability at least $1 - \delta$ starting with $\Delta = f(x^1) - f^*$:*

*Optimal tuning, independent noise:* $T = O\left(\frac{\Delta L_1}{\alpha_p^2 \varepsilon}\right), \gamma_k = \frac{\alpha_p}{4L_1}, B_k = \max\left\{1, \left(\frac{4\sigma T}{\Delta}\right)^{\frac{\kappa}{\kappa-1}}\right\}$ for $\varepsilon \geq \frac{4L_0}{L_1}$ and $T = O\left(\frac{\Delta L_0}{\alpha_p^2 \varepsilon^2}\right), \gamma_k = \sqrt{\frac{\Delta}{4TL_0}}, B_k = \max\left\{1, \left(\frac{4\sigma T}{\Delta}\right)^{\frac{\kappa}{\kappa-1}}\right\}$ for $\varepsilon \leq \frac{4L_0}{L_1}$

$$N = O\left(\frac{\Delta}{\alpha_p^2 \varepsilon}\left(L_1 + \frac{L_0}{\varepsilon}\right)\left[1 + \left(\frac{\sigma}{\alpha_p^2 \varepsilon}\left(L_1 + \frac{L_0}{\varepsilon}\right)\right)^{\frac{\kappa}{\kappa-1}}\right]\right),$$

*Optimal tuning, Lipschitz noise:* $T = O\left(\frac{\Delta L_1}{\alpha_p^2 \varepsilon}\right), \gamma_k = \frac{\alpha_p}{4L_1}, B_k = \max\left\{1, \left(\frac{32\sigma}{\alpha_p \varepsilon}\right)^{\frac{\kappa}{\kappa-1}}\right\}$ for $\varepsilon \geq \frac{4L_0}{L_1}$ and $T = O\left(\frac{\Delta L_0}{\alpha_p^2 \varepsilon^2}\right), \gamma_k = \sqrt{\frac{\Delta}{32TL_0}}, B_k = \max\left\{1, \left(\frac{32\sigma}{\alpha_p \varepsilon}\right)^{\frac{\kappa}{\kappa-1}}\right\}$ for $\varepsilon \leq \frac{4L_0}{L_1}$

$$N = O\left(\frac{\Delta}{\alpha_p^2 \varepsilon}\left(L_1 + \frac{L_0}{\varepsilon}\right)\left[1 + \left(\frac{\sigma}{\alpha_p \varepsilon}\right)^{\frac{\kappa}{\kappa-1}}\right]\right).$$

*Proof.* Consider the $k$-th step of CompSGD. We use smoothness of function $f$ (Lemma 2) to estimate:

$$
\begin{aligned}
f(x^{k+1}) - f(x^k) \quad &\leq \quad \langle \nabla f(x^k), x^{k+1} - x^k \rangle + \frac{L_0 + L_1\|\nabla f(x^k)\|_2}{2} \exp(L_1\|x^{k+1} - x^k\|_2)\|x^{k+1} - x^k\|_2^2 \\
&= \quad -\gamma_k \cdot \mathrm{sign}(f(x^k + \gamma_k \mathbf{e}^k, \xi_+^k) - f(x^k - \gamma_k \mathbf{e}^k, \xi_-^k)) \cdot \langle \nabla f(x^k), \mathbf{e}^k \rangle \\
&\quad + \quad \frac{L_0 + L_1\|\nabla f(x^k)\|_2}{2} \exp(\gamma_k L_1 \|\mathbf{e}^k\|_2)\gamma_k^2\|\mathbf{e}^k\|_2^2 \\
&\overset{As.4}{\leq} \quad -\gamma_k \frac{\mathrm{sign}(f(x^k + \gamma_k \mathbf{e}^k, \xi_+^k) - f(x^k - \gamma_k \mathbf{e}^k, \xi_-^k)) \cdot \langle \nabla f(x^k), \mathbf{e}^k \rangle}{\|\nabla f(x^k)\|_p} \cdot \|\nabla f(x^k)\|_p \\
&\quad + \quad \frac{L_0 + L_1\|\nabla f(x^k)\|_2}{2} \exp(\gamma_k L_1)\gamma_k^2.
\end{aligned}
$$

Let us choose $\gamma_k \leq \frac{1}{4L_1} \leq \frac{\alpha_p}{4L_1}$, then we have $L_1\gamma_k \exp(L_1\gamma_k) \leq \frac{\alpha_p}{4}$ and $\|\nabla f(x^k)\|_2 \leq \|\nabla f(x^k)\|_p$:

$$
\begin{aligned}
f(x^{k+1}) - f(x^k) \quad &\leq \quad -\gamma_k \frac{\mathrm{sign}(f(x^k + \gamma_k \mathbf{e}^k, \xi_+^k) - f(x^k - \gamma_k \mathbf{e}^k, \xi_-^k)) \cdot \langle \nabla f(x^k), \mathbf{e}^k \rangle}{\|\nabla f(x^k)\|_p} \cdot \|\nabla f(x^k)\|_p \\
&\quad + \quad \frac{L_0\gamma_k^2}{2} + \frac{\alpha_p\gamma_k\|\nabla f(x^k)\|_p}{8}.
\end{aligned}
\tag{35}
$$

Consequently, after summing $T$ steps, we obtain

$$
\sum_{k=1}^{T} \gamma_k \left[ \frac{\mathrm{sign}(f(x^k + \gamma_k \mathbf{e}^k, \xi_+^k) - f(x^k - \gamma_k \mathbf{e}^k, \xi_-^k)) \cdot \langle \nabla f(x^k), \mathbf{e}^k \rangle}{\|\nabla f(x^k)\|_p} - \frac{\alpha_p}{8} \right] \cdot \|\nabla f(x^k)\|_p \leq
$$

$$
\underbrace{f(x^1) - f(x^*)}_{=\Delta} + \frac{L_0}{2}\sum_{k=1}^{T}\gamma_k^2.
\tag{36}
$$

Taking math expectation from both sides, we obtain

$$
\sum_{k=1}^{T} \gamma_k \mathbb{E}_{\xi,\mathbf{e}^k}[\mathrm{sign}(f(x^k + \gamma_k \mathbf{e}^k, \xi_+^k) - f(x^k - \gamma_k \mathbf{e}^k, \xi_-^k)) \cdot \langle \nabla f(x^k), \mathbf{e}^k \rangle] - \frac{\alpha_p\gamma_k}{8}\mathbb{E}_{\xi,\mathbf{e}^k}[\|\nabla f(x^k)\|_p] \leq
$$

$$
\underbrace{f(x^1) - f(x^*)}_{=\Delta} + \frac{L_0}{2}\sum_{k=1}^{T}\gamma_k^2.
\tag{37}
$$

Next, we estimate the term

$$
\begin{aligned}
\psi_k \quad &:= \quad \mathbb{E}_{\xi,\mathbf{e}^k}\left[\mathrm{sign}(f(x^k + \gamma_k \mathbf{e}^k, \xi_+^k) - f(x^k - \gamma_k \mathbf{e}^k, \xi_-^k)) \cdot \langle \nabla f(x^k), \mathbf{e}^k \rangle\right] = \mathbb{E}_{\mathbf{e}^k}|\langle \nabla f(x^k), \mathbf{e}^k \rangle| \\
&\quad - \quad \mathbb{E}_{\mathbf{e}^k}\left[2 \cdot \mathbb{P}_\xi\left[\mathrm{sign}(f(x^k + \gamma_k \mathbf{e}^k, \xi_+^k) - f(x^k - \gamma_k \mathbf{e}^k, \xi_-^k)) \neq \mathrm{sign}(\langle \nabla f(x^k), \mathbf{e}^k \rangle)\right] \cdot |\langle \nabla f(x^k), \mathbf{e}^k \rangle|\right].
\end{aligned}
$$

We consider two cases to deal with probability over $\xi$: $|\langle \nabla f(x^k), \mathbf{e}^k \rangle| \geq 2\gamma_k L_0 + \alpha_p\|\nabla f(x^k)\|_p/8$ and $|\langle \nabla f(x^k), \mathbf{e}^k \rangle| \leq 2\gamma_k L_0 + \alpha_p\|\nabla f(x^k)\|_p/8$.

**Case $|\langle \nabla f(x^k), \mathbf{e}^k \rangle| \leq 2\gamma_k L_0 + \alpha_p\|\nabla f(x^k)\|_p/8$:**

$$
\begin{aligned}
\mathbb{E}_{\xi,\mathbf{e}^k}\quad &\left[\mathrm{sign}(f(x^k + \gamma_k \mathbf{e}^k, \xi_+^k) - f(x^k - \gamma_k \mathbf{e}^k, \xi_-^k)) \cdot \langle \nabla f(x^k), \mathbf{e}^k \rangle\right] \geq -\mathbb{E}_{\mathbf{e}^k}[|\langle \nabla f(x^k), \mathbf{e}^k \rangle|] \\
&\geq \quad \mathbb{E}_{\mathbf{e}^k}[|\langle \nabla f(x^k), \mathbf{e}^k \rangle|] - 4\gamma_k L_0 - \alpha_p\frac{\|\nabla f(x^k)\|_p}{8} \overset{As.\ 4}{\geq} \frac{7\alpha_p}{8}\|\nabla f(x^k)\|_p - 4\gamma_k L_0.
\end{aligned}
$$

**Case $|\langle \nabla f(x^k), \mathbf{e}^k \rangle| \geq 2\gamma_k L_0 + \alpha_p\|\nabla f(x^k)\|_p/8$:**

We change sign operators to equivalent ones denoting $\theta_+^k := f(x^k + \gamma_k \mathbf{e}^k, \xi_+^k) - f(x^k + \gamma_k \mathbf{e}^k)$ and $\theta_-^k := f(x^k - \gamma_k \mathbf{e}^k, \xi_-^k) - f(x^k - \gamma_k \mathbf{e}^k)$:

$$
\mathrm{sign}(f(x^k + \gamma_k \mathbf{e}^k, \xi_+^k) - f(x^k - \gamma_k \mathbf{e}^k, \xi_-^k)) \quad \neq \quad \mathrm{sign}(\langle \nabla f(x^k), \mathbf{e}^k \rangle)
$$

$$
\Updownarrow
$$

$$
\mathrm{sign}(f(x^k + \gamma_k \mathbf{e}^k) - f(x^k - \gamma_k \mathbf{e}^k) + \theta_+^k - \theta_-^k) \quad \neq \quad \mathrm{sign}(2\gamma_k \cdot \langle \nabla f(x^k), \mathbf{e}^k \rangle).
$$

Further, we can bound probability by considering larger number of cases:

$$\mathbb{P}_\xi \quad \left[ \text{sign}(f(x^k + \gamma_k \mathbf{e}^k, \xi_+^k) - f(x^k - \gamma_k \mathbf{e}^k, \xi_-^k)) \neq \text{sign}(\langle \nabla f(x^k), \mathbf{e}^k \rangle) \right] \tag{38}$$

$$= \quad \mathbb{P}_\xi \left[ \text{sign}(f(x^k + \gamma_k \mathbf{e}^k) - f(x^k - \gamma_k \mathbf{e}^k) + \theta_+^k - \theta_-^k) \neq \text{sign}(2\gamma_k \cdot \langle \nabla f(x^k), \mathbf{e}^k \rangle) \right]$$

$$\leq \quad \mathbb{P}_\xi \left[ |f(x^k + \gamma_k \mathbf{e}^k) - f(x^k - \gamma_k \mathbf{e}^k) + \theta_+^k - \theta_-^k - 2\gamma_k \cdot \langle \nabla f(x^k), \mathbf{e}^k \rangle| \geq 2\gamma_k \cdot |\langle \nabla f(x^k), \mathbf{e}^k \rangle| \right]$$

$$\leq \quad \mathbb{P}_\xi \left[ |f(x^k + \gamma_k \mathbf{e}^k) - f(x^k - \gamma_k \mathbf{e}^k) - 2\gamma_k \cdot \langle \nabla f(x^k), \mathbf{e}^k \rangle| + |\theta_+^k - \theta_-^k| \geq 2\gamma_k \cdot |\langle \nabla f(x^k), \mathbf{e}^k \rangle| \right].$$

We apply Smoothness Lemma 2 and choose $\gamma_k \leq \frac{\alpha_p}{8L_1}$ to bound the term:

$$|f(x^k + \gamma_k \mathbf{e}^k) - f(x^k - \gamma_k \mathbf{e}^k) - 2\gamma_k \cdot \langle \nabla f(x^k), \mathbf{e}^k \rangle| \quad \leq \quad 2 \cdot \frac{L_0 + L_1 \|\nabla f(x^k)\|_2}{2} \exp(L_1 \gamma_k \|\mathbf{e}^k\|_2) \gamma_k^2 \|\mathbf{e}^k\|_2^2$$

$$\leq \quad 2L_0 \gamma_k^2 + \alpha_p \|\nabla f(x^k)\|_p \gamma_k / 8.$$

We continue to estimate the probability:

$$\mathbb{P}_\xi \quad \left[ |f(x^k + \gamma_k \mathbf{e}^k) - f(x^k - \gamma_k \mathbf{e}^k) - 2\gamma_k \cdot \langle \nabla f(x^k), \mathbf{e}^k \rangle| + |\theta_+^k - \theta_-^k| \geq 2\gamma_k \cdot |\langle \nabla f(x^k), \mathbf{e}^k \rangle| \right]$$

$$\leq \quad \mathbb{P}_\xi \left[ 2L_0 \gamma_k^2 + \gamma_k \alpha_p \|\nabla f(x^k)\|_p / 4 + |\theta_+^k - \theta_-^k| \geq 2\gamma_k \cdot |\langle \nabla f(x^k), \mathbf{e}^k \rangle| \right]. \tag{39}$$

Since we consider the case $|\langle \nabla f(x^k), \mathbf{e}^k \rangle| \geq 2\gamma_k L_0 + \alpha_p \|\nabla f(x^k)\|_p / 8$, we bound

$$(19) \quad \leq \quad \mathbb{P}_\xi \left[ \gamma_k \cdot |\langle \nabla f(x^k), \mathbf{e}^k \rangle| + |\theta_+^k - \theta_-^k| \geq 2\gamma_k \cdot |\langle \nabla f(x^k), \mathbf{e}^k \rangle| \right]$$

$$\leq \quad \mathbb{P}_\xi \left[ |\theta_+^k - \theta_-^k| \geq \gamma_k \cdot |\langle \nabla f(x^k), \mathbf{e}^k \rangle| \right]$$

$$\overset{\text{Markov ineq.(12):}}{\leq} \quad \frac{\mathbb{E}_\xi[|\theta_+^k - \theta_-^k|]}{\gamma_k \cdot |\langle \nabla f(x^k), \mathbf{e}^k \rangle|}. \tag{40}$$

In case of independent noise, $\mathbb{E}_\xi[|\theta_+^k - \theta_-^k|]$ is simply bounded by the constant $\sigma_k$ and $\tilde{\sigma}_k = \sigma_k$. In case of Lipschitz noise, $\mathbb{E}_\xi[|\theta_+^k - \theta_-^k|] \leq \sigma_k \|2\gamma_k \mathbf{e}^k\|_2 = 2\sigma_k \gamma_k$ and $\tilde{\sigma}_k = 2\sigma_k \gamma_k$. Finally, we obtain the bound

$$\mathbb{E}_{\xi, \mathbf{e}^k} \left[ \text{sign}(f(x^k + \gamma_k \mathbf{e}^k, \xi_+^k) - f(x^k - \gamma_k \mathbf{e}^k, \xi_-^k)) \cdot \langle \nabla f(x^k), \mathbf{e}^k \rangle \right] \quad \geq \quad \mathbb{E}_{\mathbf{e}^k} |\langle \nabla f(x^k), \mathbf{e}^k \rangle| - \frac{4\tilde{\sigma}_k}{\gamma_k}$$

$$\overset{\text{As. 4}}{\geq} \quad \alpha_p \|\nabla f(x^k)\|_p - \frac{4\tilde{\sigma}_k}{\gamma_k}.$$

Combining two cases together, we get $\psi_k \geq \frac{7\alpha_p}{8} \|\nabla f(x^k)\|_p - 4\gamma_k L_0 - \frac{4\tilde{\sigma}_k}{\gamma_k}$ and the bound

$$\frac{1}{8} \sum_{k=1}^T \gamma_k \mathbb{E}[\|\nabla f(x^k)\|_p] \quad \leq \quad \frac{\Delta}{\alpha_p} + \frac{L_0}{2\alpha_p} \sum_{k=1}^T \gamma_k^2 + \sum_{k=1}^T \gamma_k \cdot \frac{4L_0 \gamma_k}{\alpha_p} + 4 \sum_{k=1}^T \frac{\tilde{\sigma}_k}{\alpha_p}.$$

Due to Batching Lemma 4, we can estimate the $\kappa-$th moment of the batched estimate by:

$$\sigma_k \leq \frac{2\sigma}{B_k^{\frac{\kappa-1}{\kappa}}}.$$

Plugging in constant stepsizes and batchsizes $\gamma_k \equiv \gamma$, $C_T = T\gamma^2$, $\gamma_1 = \gamma$ and dividing both sides by $\frac{\alpha_p T \gamma}{8}$ yields the bound:

$$\frac{1}{T} \sum_{k=1}^T \|\nabla f(x^k)\|_p \leq \frac{2\Delta}{T\alpha_p \gamma} + 8\frac{L_0 \gamma}{\alpha_p} + \frac{8\tilde{\sigma}}{\alpha_p \gamma B^{\frac{\kappa-1}{\kappa}}}. \tag{41}$$

Next, we pick optimal parameters for optimal convergence under independent and Lipschitz noise.

**Independent, optimal tuning,** $\varepsilon \geq \frac{4L_0}{L_1}$**:** Choosing the largest possible $\gamma = \frac{\alpha_p}{4L_1}$, we bound the term:

$$8\frac{L_0^\delta \gamma}{\alpha_p} \leq \frac{8}{4} \frac{L_0}{L_1} \leq \frac{2\varepsilon}{1}.$$

The bound (21) becomes:

$$\frac{1}{T}\sum_{k=1}^{T}\|\nabla f(x^k)\|_p \le \frac{8L_1}{\alpha_p^2}\left[\frac{\Delta}{T} + \frac{4\sigma}{B^{\frac{\kappa-1}{\kappa}}}\right] + \frac{2\varepsilon}{1}.$$

Choosing $B$ such that $\frac{4\sigma}{\Delta B^{\frac{\kappa-1}{\kappa}}} \le \frac{1}{T} \Rightarrow B = \max\left\{1, \left(\frac{4\sigma T}{\Delta}\right)^{\frac{\kappa}{\kappa-1}}\right\}$, we only need to bound

$$\frac{16L_1}{\alpha_p^2}\frac{\Delta}{T} \le \frac{\varepsilon}{2} \Rightarrow T = O\left(\frac{\Delta L_1}{\alpha_p^2 \varepsilon}\right).$$

The final sample complexity is

$$N = T \cdot B = O\left(\frac{\Delta L_1}{\alpha_p^2 \varepsilon}\left[1 + \left(\frac{\sigma L_1}{\alpha_p^2 \varepsilon}\right)^{\frac{\kappa}{\kappa-1}}\right]\right).$$

**Independent, optimal tuning, $\varepsilon \le \frac{4L_0}{L_1}$:** Choosing $B$ such that $\frac{4\sigma}{\Delta B^{\frac{\kappa-1}{\kappa}}} \le \frac{1}{T} \Rightarrow B = \max\left\{1, \left(\frac{4\sigma T}{\Delta}\right)^{\frac{\kappa}{\kappa-1}}\right\}$, we transform the bound into

$$\frac{1}{T}\sum_{k=1}^{T}\|\nabla f(x^k)\|_p \le \frac{2\Delta}{T\alpha_p\gamma} + 8\frac{L_0\gamma}{\alpha_p} \le \varepsilon.$$

Using $\gamma = \sqrt{\frac{\Delta\alpha_p}{4TL_0}}$, we obtain

$$\frac{2\Delta}{T\alpha_p\gamma} + 8\frac{L_0\gamma}{\alpha_p} = 16\sqrt{\frac{\Delta L_0}{\alpha_p^2 T}} \le \varepsilon.$$

Hence, required number of iterations $T = O\left(\frac{\Delta L_0}{\alpha_p^2 \varepsilon^2}\right)$ and the final comparison complexity is

$$N = T \cdot B = O\left(\frac{\Delta L_0}{\alpha_p^2 \varepsilon^2}\left[1 + \left(\frac{\sigma L_0}{\alpha_p^2 \varepsilon^2}\right)^{\frac{\kappa}{\kappa-1}}\right]\right).$$

We also notice that $\gamma = \sqrt{\frac{\Delta}{32TL_0}} \le \frac{\alpha_p}{4L_1}$ for this number of iterations and $\varepsilon \le \frac{4L_0}{L_1}$.

**Lipschitz, optimal tuning, $\varepsilon \ge \frac{4L_0}{L_1}$:** Choosing the largest possible $\gamma = \frac{\alpha_p}{4L_1}$, we bound the term:

$$8\frac{L_0\gamma}{\alpha_p} \le \frac{8}{4}\frac{L_0}{L_1} \le \frac{2\varepsilon}{1}.$$

The bound becomes:

$$\frac{1}{T}\sum_{k=1}^{T}\|\nabla f(x^k)\|_p \le \frac{8L_1}{\alpha_p^2}\frac{\Delta}{T} + \frac{8\sigma}{\alpha_p B^{\frac{\kappa-1}{\kappa}}} + \frac{2\varepsilon}{1}.$$

Choosing $B$ such that $\frac{8\sigma}{\alpha_p B^{\frac{\kappa-1}{\kappa}}} \le \varepsilon/4 \Rightarrow B = \max\left\{1, \left(\frac{32\sigma}{\alpha_p \varepsilon}\right)^{\frac{\kappa}{\kappa-1}}\right\}$, we only need to bound

$$\frac{8L_1}{\alpha_p^2}\frac{\Delta}{T} \le \frac{\varepsilon}{4} \Rightarrow T = O\left(\frac{\Delta L_1}{\alpha_p^2 \varepsilon}\right).$$

The final sample complexity is

$$N = T \cdot B = O\left(\frac{\Delta L_1}{\alpha_p^2 \varepsilon}\left[1 + \left(\frac{\sigma}{\alpha_p \varepsilon}\right)^{\frac{\kappa}{\kappa-1}}\right]\right).$$

**Lipschitz, optimal tuning,** $\varepsilon \leq \frac{4L_0}{L_1}$**:** Choosing $B$ such that $\frac{4\sigma}{\alpha_p B^{\frac{\kappa-1}{\kappa}}} \leq \varepsilon/4 \Rightarrow B = \max\left\{1, \left(\frac{16\sigma}{\alpha_p \varepsilon}\right)^{\frac{\kappa}{\kappa-1}}\right\}$, we transform the bound (22) into

$$\frac{1}{T}\sum_{k=1}^{T} \|\nabla f(x^k)\|_p \leq \frac{2\Delta}{T\alpha_p\gamma} + 8\frac{L_0\gamma}{\alpha_p} \leq \varepsilon.$$

Using $\gamma = \sqrt{\frac{\Delta}{16TL_0}}$, we obtain

$$\frac{2\Delta}{T\alpha_p\gamma} + 8\frac{L_0\gamma}{\alpha_p} = 16\sqrt{\frac{\Delta L_0}{\alpha_p^2 T}} \leq \varepsilon.$$

Hence, required number of iterations $T = O\left(\frac{\Delta L_0}{\alpha_p^2\varepsilon^2}\right)$ and the final comparison complexity is

$$N = T \cdot B = O\left(\frac{\Delta L_0}{\alpha_p^2\varepsilon^2}\left[1 + \left(\frac{\sigma}{\alpha_p\varepsilon}\right)^{\frac{\kappa}{\kappa-1}}\right]\right).$$

We also notice that $\gamma = \sqrt{\frac{\Delta}{32TL_0}} \leq \frac{\alpha_p}{48L_1}$ for this number of iterations and $\varepsilon \leq \frac{4L_0}{L_1}$. $\qquad\square$

### A.9 IN EXPECTATION CONVERGENCE OF MajorityVote-CompSGD

**Theorem 8 (In expectation complexity for MajorityVote-CompSGD).** *Consider lower-bounded $(L_0, L_1)$-smooth function $f$ (As. 1, 2), random directions with $\alpha_p$ (As. 4) and function estimates with HT **independent, unimodal and symmetric** noise $\kappa > 0$ (As. 3). Then Alg. 2 requires comparison number $N$ to achieve $\frac{1}{T}\sum_{k=1}^{T}\mathbb{E}[\|\nabla f(x_k)\|_p] \leq \varepsilon$ staring with $\Delta = f(x^1) - f^*$:*

***Optimal tuning, independent noise:*** $T = O\left(\frac{\Delta L_1}{\alpha_p^2\varepsilon}\right), \gamma_k = \frac{\alpha_p}{4L_1}, M_k = \max\left\{\frac{160}{\kappa^2}, \left(\frac{4\sigma T}{\Delta}\right)^2\right\}$ *for*

$\varepsilon \geq \frac{4L_0}{L_1}$ *and* $T = O\left(\frac{\Delta L_0}{\alpha_p^2\varepsilon^2}\right), \gamma_k = \sqrt{\frac{\Delta}{4TL_0}}, M_k = \max\left\{\frac{160}{\kappa^2}, \left(\frac{4\sigma T}{\Delta}\right)^2\right\}$ *for* $\varepsilon \leq \frac{4L_0}{L_1}$

$$N = O\left(\frac{\Delta}{\alpha_p^2\varepsilon}\left(L_1 + \frac{L_0}{\varepsilon}\right)\left[\frac{1}{\kappa^2} + \left(\frac{\sigma}{\alpha_p^2\varepsilon}\left(L_1 + \frac{L_0}{\varepsilon}\right)\right)^2\right]\right),$$

***Optimal tuning, Lipschitz noise:*** $T = O\left(\frac{\Delta L_1}{\alpha_p^2\varepsilon}\right), \gamma_k = \frac{\alpha_p}{4L_1}, M_k = \max\left\{\frac{160}{\kappa^2}, \left(\frac{32\sigma}{\alpha_p\varepsilon}\right)^2\right\}$ *for*

$\varepsilon \geq \frac{4L_0}{L_1}$ *and* $T = O\left(\frac{\Delta L_0}{\alpha_p^2\varepsilon^2}\right), \gamma_k = \sqrt{\frac{\Delta}{32TL_0}}, M_k = \max\left\{\frac{160}{\kappa^2}, \left(\frac{32\sigma}{\alpha_p\varepsilon}\right)^2\right\}$ *for* $\varepsilon \leq \frac{4L_0}{L_1}$

$$N = O\left(\frac{\Delta}{\alpha_p^2\varepsilon}\left(L_1 + \frac{L_0}{\varepsilon}\right)\left[1 + \left(\frac{\sigma}{\alpha_p\varepsilon}\right)^2\right]\right).$$

The proof completely copies the proof of in expectation convergence Theorem 7 for minibatch-CompSGD under $\kappa = 2$ combined with MajorityVote-CompSGD Convergence Lemma 5.

## B EXPERIMENTAL DETAILS

### B.1 ROBERTA LARGE FINE-TUNING

For these experiments, we follow (Gao et al., 2020b) for the prompt-based fine-tuning paradigm for masked language models and reuse training hyperparameters from (Malladi et al., 2023a). Please refer to the original papers for more details. We compare methods in few-shot scenario with $k = 16$ examples.

For minibatch-CompSGD Algorithm 3, we sampled $\mathbf{e}^k$ from scaled Euclidian sphere, i.e. $\alpha \cdot S_2^d = \{\mathbf{e}|\|\mathbf{e}\|_2 = \alpha\}$. We set $\alpha$ equal to 17 for all datasets and selected the learning rate in [0.3, 1.0, 3.0] based on validation score.

## C  PROMPTS

Below we present the prompts used in our experiments.

---

**Prompt for human face generation for diffusion**

Ultra-realistic portrait of a person, highly detailed facial features, natural lighting, skin texture visible, professional studio quality, 4K resolution

---

**Prompt to get a description of a picture of a human face**

Act as an expert forensic facial analyst. Provide a highly detailed, objective, and technical analysis of the facial features in the provided image. Focus only on observable visual characteristics.
Analyze the following key traits comprehensively:

- **Eye Structure:** Describe shape, explicitly state perceived eye color (is it Green?) and intensity, iris patterns, interocular distance.
- **Skin Topography:** Detail texture, color/tone.
- **Freckles:** Note presence, density, and distribution of distinct freckles.
- **Scars:** Crucially, note any visible scars on the face – location, shape, appearance.
- Detail wrinkles/rhytids, specifically noting the presence, pattern (e.g., horizontal, vertical), and apparent depth of any forehead wrinkles (rhytids on the frontal region), and moles.

**Hair Analysis:**

- **Eyebrows:** Describe shape, thickness, density. Are they bushy or sparse?
- **Head Hair:** Color, texture, hairline, density, and length (e.g., short/cropped, above shoulders, shoulder-length, below shoulders, significantly long).

**Facial Bone Structure:**

- **Cheekbones:** Prominence. Are they high/prominent or less defined?
- **Jaw and Chin Structure:** Jawline shape. Chin shape and explicitly state if a cleft chin is present or absent.
- **Nasal Structure:** Bridge shape, nostril shape, size. Specifically comment if the nose has a Roman profile.

**Lip and Philtrum Morphology:**

- Describe shape and relative fullness of upper/lower lips.
- Crucially, examine the upper lip and philtrum (the groove between the base of the nose and the upper lip). Is there any visible indication of a cleft lip (also known as 'harelip' or 'cheiloschisis')? This could be a repaired scar, an indentation, or an asymmetry in the lip or nostril base associated with a cleft. Describe any such findings.

**Background Characteristics:** Describe the background. Is it neutral (e.g., plain, blurred, studio-like, uniform color) or does it provide discernible environmental context (e.g., outdoor scene, specific room details)?
Include these specific assessments:

- Potential Ancestry Indicators (objective).
- Facial Symmetry (brief).
- Age-Related Indicators.

**Image Realism Assessment:** Describe the perceived level of realism of the image. Is it photorealistic, CGI-generated, an artistic rendering, a painting, heavily edited, etc.? Note any visual cues supporting this assessment (e.g., unnatural lighting, skin texture, artifacts).
**Output Format:** Produce a technical description. Use precise terms. Avoid subjective interpretations. Ensure the points about Green Eyes, Visible Scars, Cleft Chin, High/Prominent Cheekbones, Bushy Eyebrows, Distinct Freckles, Roman Nose, Cleft Lip, forehead wrinkles, and hair length are clearly addressed, even if to state their absence.

---

## Prompt to evaluate two descriptions and select a more relevant person

**[SYSTEM ROLE]** You are a precise and objective Evaluator. Your task is to analyze two facial descriptions based only on a specific set of criteria and determine which description better meets them. You MUST ignore any information not relevant to the specific criteria.

**[TASK]** Evaluate the two descriptions below (**[DESCRIPTION 1]** and **DESCRIPTION 2**) based on the **[SCORING LOGIC]**. Award points strictly based on whether each feature's presence (as specified) is explicitly stated in the text. Then, apply tie-breaking rules if necessary.

**[PRIMARY CRITERIA]** The evaluation focuses on two features:

- Presence of Distinct Freckles

- Presence of Green Eyes (including clear shades of green). **Important Note:** If the description indicates that the eyes appear green only because the entire image or scene has an overall green tint, filter, or color cast (e.g., "the whole picture is green, making the eyes look green"), this does not count as true green eyes, and 0 points should be awarded for Green Eyes in such cases. The green color must be inherent to the eyes themselves.

**[SCORING LOGIC]** Points are awarded based on the explicit statement of features as follows:

**Freckles Score** (for each description):

- Award +50 points if the description explicitly states that distinct freckles are visible or present.

- A score of 0 points for freckles is given if the description explicitly states their absence (e.g., "no freckles," "freckles are absent") or if freckles are not mentioned at all.

**Green Eyes Score** (for each description):

- Award +50 points to a description if it explicitly states the presence of "green eyes", "vivid green eyes", or any phrase clearly indicating the eyes are a shade of green (e.g., "light green eyes", "dark green eyes", "emerald eyes", "olive green eyes", "sea-green eyes"), AND this green color is attributed to the eyes themselves, not to an overall image tint.

- A score of 0 points for green eyes is given if such green eyes (or shades of green) are not mentioned, if a different eye color is stated, or if the description indicates the eyes appear green solely due to an overall green tint/filter/color cast on the image/scene.

**Total Score** (for each description):

- The sum of its "Freckles Score" and "Green Eyes Score".

- Maximum possible total score per description: 100 points.

**[TIE-BREAKING CRITERIA - Apply in order if total scores are equal]**
**Desired Background:**

- A "Desired Background" is one that is described as neutral (e.g., plain, blurred, studio-like, a uniform color, or lacking specific environmental details that allow identification of a real-world location) AND is described without any indication of strong, artificial, or unnatural color palette shifts across the background. The background palette should seem natural or unmanipulated.

- Any other type of background is considered "Not Desired". This includes:
    - Contextual backgrounds (showing discernible environmental elements like an outdoor scene, specific room details, identifiable objects).
    - Neutral backgrounds where the description does suggest strong, artificial, or unnatural color palette shifts (e.g., "plain background with an unnatural green tint over everything", "studio background with oversaturated neon colors").
    - Backgrounds where neutrality or palette cannot be determined from the description to meet the "Desired" criteria.

If total scores from the **[SCORING LOGIC]** are equal, prefer the description that more clearly indicates a Desired Background.

**Default Selection:** If total scores from the **[SCORING LOGIC]** are equal AND the Desired Background assessment does not result in a clear preference for one description (e.g., both are Desired, both are Not Desired, or it's impossible to distinguish based on the provided text to give one a clear advantage), select Description 1.

**[DATA]**
**[DESCRIPTION 1]**: {description_1}
**[DESCRIPTION 2]**: {description_2}
**[INSTRUCTIONS]** Carefully read both descriptions.

- For each description, calculate its Freckles Score (+50 or +0) according to the **[SCORING LOGIC]**.

- Calculate the Green Eyes Score (+50 or +0) for each description according to the **[SCORING LOGIC]**, paying close attention to the rule about overall image greenness.

- Calculate the Total Score for each description (Freckles Score + Green Eyes Score).

**Selection Process:**

    a. If the Total Scores are unequal, select the description with the higher Total Score.

    b. If the Total Scores are equal:

        i. Evaluate **[DESCRIPTION 1]** and **DESCRIPTION 2** based on the Desired Background tie-breaking criterion.

        ii. If one description indicates a "Desired Background" and the other does not (is "Not Desired" or unclear such that it cannot be confirmed as "Desired"), select the description with the "Desired Background".

        iii. If the Desired Background criterion is inconclusive (e.g., both are clearly "Desired", both are clearly "Not Desired", or information is insufficient to make a distinction), proceed to the Default Selection tie-breaker (select Description 1).

Format your response exactly as specified below. Do not add any extra text before or after.

**[OUTPUT FORMAT]** Provide your response STRICTLY in the following format:

Line 1: `[Y]` where Y is the number (1 or 2) of the selected description.

Line 2: `X/100` where X is the total score (from the **[SCORING LOGIC]**) you calculated for the selected description.

Line 3: `Reason:` followed by a concise explanation.

- Start by stating the scores for both descriptions. For Freckles, state the score. For Green Eyes, state the Green Eyes score. Finally, state their total scores. (e.g., "Desc 1: Freckles +50, Green Eyes +50. Total: 100. Desc 2: Freckles +50, Green Eyes +0. Total: 50." OR "Desc 1: Freckles +0, Green Eyes +50. Total: 50. Desc 2: Freckles +0, Green Eyes +0. Total: 0.")

- If total scores were unequal, explain why the selected description was chosen based on its higher total score.

- If a tie-breaker was used (because total scores were equal), explicitly state which tie-breaker (**[Desired Background]** or **[Default Selection]**) was applied and why.

- If **[Desired Background]** was applied and led to a selection, briefly describe the background assessment for both descriptions (e.g., "Background: Desired," "Background: Not Desired," "Background: Unclear") that led to the choice.

- If **[Desired Background]** was inconclusive, state this and explain why (e.g., "Both backgrounds Desired," "Both backgrounds Not Desired," "Backgrounds unclear for distinction"), then state that **[Default Selection]** was applied.

**[EXAMPLE 1 - Unequal Total Scores]**

Assume:

Description 1: "The person has many distinct freckles. Their eyes are a vivid green. Background is a plain white wall."

Description 2: "Distinct freckles cover their nose. Eyes are blue. Background is a busy street."

Output for this example:

```
[1]
100/100
Reason:  Desc 1:  Freckles +50, Green Eyes +50.  Total:  100.  Desc
2:  Freckles +50, Green Eyes +0.  Total:  50.  Selected 1 due to
higher total score.
```

**[EXAMPLE 2 - Equal Total Scores; Tie-breaker: Desired Background leads to selection (green tint issue in one description)]**

Assume:

Description 1: "Distinct freckles are visible. Her eyes appear green, but this is because the entire photograph has a heavy green tint over it. The background is a detailed outdoor park."

Description 2: "No freckles are present. The person has vivid emerald green eyes. The background is blurred."

Output for this example:

```
[2]
50/100
Reason:  Desc 1:  Freckles +50, Green Eyes +0 (eyes green due to
overall tint).  Total:  50.  Desc 2:  Freckles +0, Green Eyes +50.
Total:  50.  Total scores equal.  Tie-breaker [Desired Background]
applied.  Desc 1 background:  contextual (outdoor park) - Not
```

```
Desired.  Desc 2 background:  neutral (blurred) - Desired.  Selected
Desc 2 for Desired Background.
```
**[EXAMPLE 3 - Equal Total Scores; Tie-breaker: Desired Background leads to selection]**
Assume:
Description 1: "Distinct freckles are present. Eyes are a striking green. The setting is an outdoor park."
Description 2: "The person has distinct freckles and vivid green eyes. The background is a blurred, uniform grey studio backdrop."
Output for this example:
```
[2]
100/100
Reason:  Desc 1:  Freckles +50, Green Eyes +50.  Total:  100.  Desc
2:  Freckles +50, Green Eyes +50.  Total:  100.  Total scores
equal.  Tie-breaker [Desired Background] applied.  Desc 1 background:
contextual (outdoor park) - Not Desired.  Desc 2 background:
neutral, natural palette (blurred, uniform grey studio backdrop) -
Desired.  Selected Desc 2 for Desired Background.
```
**[EXAMPLE 4 - Equal Total Scores; Tie-breaker: Default Selection]**
Assume:
Description 1: "The individual has distinct freckles and brown eyes. The background is a city street."
Description 2: "Distinct freckles are noticeable. They have blue eyes. The background shows an office interior."
Output for this example:
```
[1]
50/100
Reason:  Desc 1:  Freckles +50, Green Eyes +0.  Total:  50.  Desc
2:  Freckles +50, Green Eyes +0.  Total:  50.  Total scores
equal.  Tie-breaker [Desired Background] was inconclusive.  Desc
1 background:  contextual (city street) - Not Desired.  Desc 2
background:  contextual (office interior) - Not Desired.  Both
backgrounds Not Desired.  Tie-breaker [Default Selection] applied,
selecting Description 1.
```
**[EXAMPLE 5 - Equal Total Scores (all 0); Tie-breaker: Default Selection (BG inconclusive)]**
Assume:
Description 1: "The individual has clear skin, no freckles. Eyes are blue." (No background info)
Description 2: "No freckles observed. Eyes are dark. The park behind her is lovely."
Output for this example:
```
[1]
0/100
Reason:  Desc 1:  Freckles +0, Green Eyes +0.  Total:  0.  Desc
2:  Freckles +0, Green Eyes +0.  Total:  0.  Total scores equal.
Tie-breaker [Desired Background] was inconclusive.  Desc 1
background:  Unclear.  Desc 2 background:  contextual (park) - Not
Desired.  Backgrounds unclear for distinction.  Tie-breaker [Default
Selection] applied, selecting Description 1.
```
**[EXAMPLE 6 - Green Eyes vs Green Tint, Unequal Scores]**
Assume:
Description 1: "Distinct freckles. Her eyes are truly emerald green. Background is a plain studio wall."
Description 2: "Distinct freckles. The whole image is bathed in a green light, making her eyes appear green, though their true color isn't stated. Background is a simple, blurred outdoor bokeh with this green cast."
Output for this example:
```
[1]
100/100
Reason:  Desc 1:  Freckles +50, Green Eyes +50.  Total:  100.  Desc
2:  Freckles +50, Green Eyes +0 (green due to overall image tint).
Total:  50.  Selected 1 due to higher total score.
```
Now, perform the evaluation on the provided descriptions.

---

**Prompt to evaluate a single description**

**[TASK DEFINITION]** Your SOLE task is to evaluate the provided INPUT DESCRIPTION text based STRICTLY on two criteria: "Presence of Distinct Freckles" and "Presence of Green Eyes". You MUST output ONLY a single integer representing the total score (0, 1, or 2).
**[CRITERIA AND SCORING LOGIC]**
**Freckles Score:**

- Award +1 point if the INPUT DESCRIPTION explicitly states that distinct freckles are visible or present.
- Award 0 points if distinct freckles are explicitly stated as absent or are not mentioned at all.

**Green Eyes Score:**

- Award +1 point if the INPUT DESCRIPTION explicitly states the presence of "green eyes", "vivid green eyes", or any phrase clearly indicating the eyes are a shade of green (e.g., "light green eyes", "dark green eyes", "emerald eyes", "olive green eyes", "sea-green eyes"), AND this green color is attributed to the eyes themselves and NOT solely due to an overall image green tint/filter.
- Award 0 points if green eyes (or qualifying shades) are not mentioned, if a different eye color is stated, or if the description indicates the eyes appear green SOLELY due to an overall green tint/filter/color cast on the image/scene (e.g., "the whole picture is green, making the eyes look green" - this scores 0 for Green Eyes).

**Total Score:**

- The sum of "Freckles Score" and "Green Eyes Score".
- Maximum possible total score: 2.

**[INPUT DESCRIPTION]**: `{description}`
**[OUTPUT REQUIREMENT - CRITICAL]** You MUST respond with ONLY a single integer representing the Total Score (0, 1, or 2). DO NOT include ANY other words, explanations, introductory phrases, or conversational text. JUST the number.
**[EXAMPLES OF CORRECT OUTPUT FORMAT]**

- If INPUT DESCRIPTION implies Freckles Score +1 and Green Eyes Score +1, your output is: `2`
- If INPUT DESCRIPTION implies Freckles Score +1 and Green Eyes Score +0, your output is: `1`
- If INPUT DESCRIPTION implies Freckles Score +0 and Green Eyes Score +1, your output is: `1`
- If INPUT DESCRIPTION implies Freckles Score +0 and Green Eyes Score +0, your output is: `0`
- If INPUT DESCRIPTION states "Distinct freckles. Eyes appear green due to an overall green filter." (Freckles +1, Green Eyes +0), your output is: `1`

Based on the INPUT DESCRIPTION provided above, calculate the Total Score according to the **[CRITERIA AND SCORING LOGIC]** and output ONLY the resulting integer.

## D  THE USE OF LARGE LANGUAGE MODELS (LLMS)

Large Language Models were used only to check and correct grammar, as well as to rephrase short parts of the text for improved clarity.

