# OpenReview forum: "CompSGD: Robust Comparison-Based Approach for Zeroth-Order Optimization under $(L_0, L_1)$-Smoothness and Heavy-Tailed Noise"
_ICLR.cc/2026/Conference — Submitted to ICLR 2026_

### Official Review · Reviewer_58Dz · 2025-10-28

**Soundness:** 3
**Presentation:** 2
**Contribution:** 3
**Rating:** 6
**Confidence:** 4

**Summary:**

The paper studies non-convex stochastic optimization when gradients are unavailable and function values can be noisy. It focuses on zeroth-order settings where the algorithm can access either noisy function values or only pairwise comparisons indicating which of two points has a larger value. Motivated by practical scenarios like human-in-the-loop preference learning and complex deep models, the authors depart from classical smoothness and light-tailed noise assumptions. They adopt generalized $(L_0, L_1)%-smoothness, which captures cases where Hessian norms grow with the gradient, and model noise that is heavy-tailed with only bounded moments.

**Strengths:**

- The paper tackles an underexplored setting by analyzing CompSGD and variants under $(L_0, L_1)$-smoothness and heavy-tailed noise models.

- The paper provides theoretical results, namely high-probability complexity bounds for comparison-only oracles under heavy-tailed noise and $(L_0, L_1)$-smoothness.

- The main theorems are stated with explicit dependencies on the parameters.

- The provided bounds track key problem parameters (dimension, smoothness constants $L_0$ and $L_1$, noise level, target accuracy), providing the trade-offs.

- The paper motivates $(L_0, L_1)$-smoothness with relevant citations and situates the work within references where comparison oracles are natural.

**Weaknesses:**

- How tight are the bounds you provide? Can you also provide lower bounds under $(L_0, L_1)$-smoothness?

- For a general black-box optimization problem, how can one check when standard L-smoothness fails but $(L_0, L_1)$ holds?

- In the numerical tests, what are the implementation details: direction sampling, mini batch, step sizes...

- The numerical experiments need more diverse and challenging experiments. For instance, experiment with controlled synthetic data sweeping $\kappa$ from 2 to close to 1 with both symmetric and biased noise to confirm the dependence of the bounds on this parameter...

- The paper needs improvement for the writing, for instance, starting from the abstract, the authors use several constants in the bounds that they introduced later.

- Second bound in the abstract is not good, take for instance $\kappa =1$, the bound gives +\infty which is a trivial bound…

- CompSGD already exists; however, from the paper, it looks like it is a new algorithm proposed by the authors.

- Flaws, typos: some examples:
-line 224 just before Algo 1 statement “$x^{k+1} = x^k -\gamma_k e^k$, change to  $x^{k+1} = x^k + \gamma_k e^k$
-page 5, line 261 “the explicit function values with are not available” change to “the explicit function values are not available”

**Questions:**

- How tight are the bounds you provide? Can you also provide lower bounds under $(L_0, L_1)$-smoothness?

- For a general black-box optimization problem, how can one check when standard L-smoothness fails but $(L_0, L_1)$ holds?

- In the numerical tests, what are the implementation details: direction sampling, mini batch, step sizes...

- The numerical experiments need more diverse and challenging experiments. For instance, experiment with controlled synthetic data sweeping $\kappa$ from 2 to close to 1 with both symmetric and biased noise to confirm the dependence of the bounds on this parameter...

- The paper needs improvement for the writing, for instance, starting from the abstract, the authors use several constants in the bounds that they introduced later.

- Second bound in the abstract is not good, take for instance $\kappa =1$, the bound gives +\infty which is a trivial bound…

- CompSGD already exists; however, from the paper, it looks like it is a new algorithm proposed by the authors.

- Flaws, typos: some examples:
-line 224 just before Algo 1 statement “$x^{k+1} = x^k -\gamma_k e^k$, change to  $x^{k+1} = x^k + \gamma_k e^k$
-page 5, line 261 “the explicit function values with are not available” change to “the explicit function values are not available”

---

### Official Review · Reviewer_Ej2M · 2025-10-31

**Soundness:** 3
**Presentation:** 3
**Contribution:** 3
**Rating:** 4
**Confidence:** 2

**Summary:**

The paper studies nonconvex zeroth-order optimization when (i) only function comparisons (dueling/sign feedback) or function values are available, (ii) the objective satisfies generalized $(L_0, L_1)$-smoothness, and (iii) the noise is heavy-tailed with only a bounded kth moment. It proposes two sign-based algorithms built around a comparison step: CompSGD: a backbone comparison-only stochastic descent, MajorityVote-CompSGD: aggregates multiple noisy comparison signs via majority vote (comparison oracle), minibatch-CompSGD: for the value oracle, batching reduces heavy-tailed noise. Main results include high-probability (HP) convergence rates for nonconvex problems under  $(L_0, L_1)$-smoothness and heavy-tailed independent or Lipschitz noise, plus expectation bounds with milder d-dependence. Experiments verifying the results on fine tuning are provided.

**Strengths:**

- Prior nonconvex ZO results typically assume standard $(L_0, L_1)$ -smoothness and bounded variance; here, the authors handle generalized smoothness and heavy-tailed independent or Lipschitz noise and still obtain HP bounds. This is an interesting theoretical finding and makes the comparison-only case more than a heuristic: MajorityVote-CompSGD attains an HP complexity that is explicit in $\Delta, L_0, L_1, d, \kappa, \sigma$. with expectation variants showing milder d-dependence (and optimality under deterministic standard smoothness)
- The paper cleanly separates what can be done without explicit function values (aggregate signs via majority vote) versus what could be done with values (minibatch to tame tails). The MajorityVote construction makes sense for dueling feedback, and the authors explicitly call out the extra assumption needed for vote concentration
- By distinguishing regimes where $L_1$ governs step-size ceilings versus regimes where small $L_0$ permits larger, near-constant steps, the theory explains the “fast then slow” rate transition and gives practical guidance (e.g., when larger steps are justified). The explicit HP thresholds on $\sigma$ for independent vs. Lipschitz noise further illuminate when normalization-style robustness is enough and when noise must shrink with $\epsilon$

**Weaknesses:**

- HP d-dependence looks too pessimistic for high-dimensional practice. The standout caveat is the factor $\tilde{O}(d^\frac{9}{2})$ in the HP bounds (Euclidean-sphere directions). While the authors note the expectation bounds have much milder d (and match classical optimal rates in the deterministic standard-smoothness case), the headline contribution is HP. It would be interesting to see (a) a tighter analysis for structured/coordinate directions, preconditioning, or mixed direction sets that improve $\alpha_p$, or (b) empirical evidence that wall-clock/query growth scales closer to the expectation-style d in realistic tasks.
- Majority voting requires per-trial error < 1/2 and can fail for skewed or multi-modal sign noise (e.g., human/LLM judges with bias or context drift). This is a strong shape assumption on the sign noise distribution; in open-loop LLM-as-a-judge settings, stationarity and symmetry are questionable. The paper would benefit from a fallback aggregation (median-of-means or tournament-style) with guarantees under weaker assumptions, or at least diagnostics to detect violations in practice.
- The experiments should mirror that with budget-vs-accuracy curves (comparisons, value calls, wall-clock time) and multi-seed CIs. For comparison-only tasks, an ablation over vote size M and tail index $\kappa$ would connect practice to theory. A small-to-large dimension plot (and direction-set ablations: sphere vs. coordinate vs. structured) would be interesting to address the $d^\frac{9}{2}$ factor.

**Questions:**

- What happens when majority-vote assumptions are violated (skewed, drifting, or correlated sign noise)? Given your own caution that majority voting may not improve error for skewed distributions, can you: (a) add a diagnostic that flags when per-trial sign error likely exceeds 1/2, (b) implement a robust aggregator (median-of-means over groups of signs, or tournament selection) and state an HP result under weaker assumptions, or (c) bound the performance degradation when the vote condition is marginally violated?
- Can you reduce the HP d-exponent via geometry or preconditioning—and show it empirically? Your discussion notes sphere directions are preferable because $\alpha_p$ dominates, but that choice drives the $d^\frac{9}{2}$ bounds in high probability. Could coordinate, block-coordinate, or preconditioned direction sets improve the exponent while keeping comparison-only updates?

---

### Official Review · Reviewer_PhJn · 2025-11-01

**Soundness:** 2
**Presentation:** 3
**Contribution:** 3
**Rating:** 4
**Confidence:** 2

**Summary:**

The paper introduces CompSGD, a comparison-based zeroth-order method, and two robust variants: MajorityVote-CompSGD (uses repeated comparisons with majority voting) and minibatchCompSGD (batches function values when they are available). The theory is developed under generalized $(L_0, L_1)$-smoothness with heavy-tailed (bounded $\kappa$-th moment) noise, covering both independent and Lipschitz noise models.

**Strengths:**

The work cleanly states oracle models, ( $L_0, L_1$ )-smoothness, and random-direction assumptions, proves a backbone CompSGD lemma, and then layers robustification on topmaking the logic easy to follow and reuse.

**Weaknesses:**

**Question 1:**  This paper requires preference-type inputs while assuming a symmetric noise distribution. This effectively imposes a very strong condition on the expectation—one that is rarely satisfied in practice. For example, if the input encodes a 0/1 preference and the noise is fixed, the setup essentially forces the expected preference to be 0.5, which is unrealistic in most real-world scenarios.

**Question 2:** This paper does not provide a systematic analysis of the complexity overhead of MajorityVote (or median-based estimators) relative to simple averaging, particularly under non-heavy-tailed (light-tailed) noise. The analysis of Algorithm 2 (MajorityVote-CompSGD) is carried out entirely in the heavy-tailed regime and does not quantify, for the light-tailed case, how many additional samples a median-based procedure requires to match the variance of averaging. As a result, under light-tailed noise the median approach typically incurs larger constants and may converge more slowly. Without a principled analysis of this regime, readers cannot reliably assess the algorithm’s computational cost.

**Question 3:** For the experimental section, the evidence is relatively thin. The main shortcomings are twofold:
(1) The proposed CompSGD involves many hyperparameters, and a sensitivity analysis is essential for evaluating the method’s robustness and generality.
(2) The range of evaluation settings is limited—only one synthetic experiment and one extension. While this is not fatal for an optimization paper, adding one or two additional application scenarios would substantially strengthen the work’s impact (for example, RLHF), especially given that the method appears broadly applicable.

**Questions:**

See the weakness.  I will be willing to re-evaluate my score contingent on satisfactory clarifications.

---

### Official Review · Reviewer_SvV1 · 2025-11-08

**Soundness:** 2
**Presentation:** 2
**Contribution:** 2
**Rating:** 2
**Confidence:** 2

**Summary:**

The paper proposes two zero-order optimization algorithms based on CompSGD, which samples a random direction and estimates its sign to optimize the objective via a zeroth-order oracle. It proposes MajorityVote-CompSGD, which, for a minibatch, computes the sign via a zeroth-order oracle for each sample, and chooses the sign according to the majority. The second proposal is minibatch-CompSGD, which applies the zeroth-order oracle on the average of the stochastic functions of the minibatch. The paper provides theoretical guarantees using a more general $(L_0, L_1)$-smoothness assumption, as well as heavy-tailed noise with a bounded $\kappa$-th moment. It also provides numerical validation for few-shot fine-tuning of RoBERTa-large on NLP classification tasks and for fine-tuning a diffusion model for image generation.

**Strengths:**

* The paper presents new theoretical results for zero-order methods under a more general assumption of $(L_0, L_1)$-smoothness.
* The proposed algorithms are clear, simple, and computationally efficient.
* The empirical experiments are conducted across diverse tasks and demonstrate improvements over existing baselines.

**Weaknesses:**

* To achieve the reported theoretical guarantees, both Theorem 1 and Theorem 2 (for independent noise) employ huge batch sizes of $T^2$ (for $\kappa=2$).
* Following the previous point, the theoretical results may not be equivalent to those obtained under the standard smoothness assumption with bounded variance, as discussed in the related work section (when ignoring the additional $d$ factor, which may also be significant).
Could the authors specify what the parameter choices of the previous works were under the standard $L$-smooth assumption?
* The experimental results appear to be single runs without seed averaging or error bars, which is a standard in the ML community. Additionally, given the randomized nature of the proposed methods, reporting error bars across multiple seeds may be even more meaningful.
* The authors did not provide any supplementary material. Since no code for the empirical evaluation is available, the community cannot verify or assess the proposed approach's contribution.

**Questions:**

The proposed algorithms move along a random direction, with only the sign being controlled. How sensitive are they to randomizations? Additionally, did the authors attempt to compare their results with standard first-order baselines, such as SGD and Sign-SGD? These would provide a strong baseline to understand the performance of the zeroth-order approaches.

---

### Official Review · Reviewer_xeBx · 2025-11-10

**Soundness:** 3
**Presentation:** 3
**Contribution:** 2
**Rating:** 4
**Confidence:** 4

**Summary:**

Authors propose new zeroth-order methods to deal with generalized (L0,L1)-smoothness and severe heavy-tailed noise with bounded κ-th moment. Using only comparisons of function values at two different points, our MajorityVote-CompSGD method　achieves the first-known high probability bound．If function　values are available, our minibatch-CompSGD can converge to the desired average　gradient norm．In addition, authors provide convergence guarantees for Lipschitz noise,　parameter-free tunings and in expectation bounds with milder d dependence.

**Strengths:**

This paper provides a comprehensive analysis：
1. Authors propose our robust MajorityVote-CompSGD (Algorithm 2) that uses only function comparisons for optimization. To achieve accuracy ε in average ℓ2-gradient norm.
2. For zeroth-order oracle where function values are available, authors present our minibatch-CompSGD (Algorithm 3) with any independent noise  complexity (Theorem 2)
3. authors provide convergence guarantees for Lipschitz noise, parameter-free algorithms tunings and in expectation bounds with milder d dependence in the corresponding theorems.

**Weaknesses:**

1. Authors claims to have proposed a parameter-free algorithm, yet in the step size setting on lines 1216，1140，1167　 $\mathcal{r}_k=\mathcal{r}_0\leq\frac{\alpha^2_p}{48L^{\delta,p}_1}$, it relies on the parameter L1. Since parameter-free is independent of the smoothness coefficient （L0，L1）, the term "parameter-free" is not entirely accurate．
2. It is best to write the symbol f (x, ξ x) as f(x ；ξx)
3. It seems that assumption 4 is relatively strong.
4. In the experimental section, there are only two comparison algorithms, it is best to add some mainstream comparison algorithms from the past two years.

**Questions:**

1. Authors claims to have proposed a parameter-free algorithm, yet in the step size setting on lines 1216，1140，1167　 $\mathcal{r}_k=\mathcal{r}_0\leq\frac{\alpha^2_p}{48L^{\delta,p}_1}$, it relies on the parameter L1. Since parameter-free is independent of the smoothness coefficient （L0，L1）, the term "parameter-free" is not entirely accurate．
2. It seems that assumption 4 is relatively strong. Is there  always  a constant \alpha_p that satisfies $\alpha_p\in (0,1]$.

---

### Meta-Review · Area_Chair_tMwf · 2025-12-15

**Summary:**

The reviewers agree that the paper provides a comprehensive analysis of comparison-based methods under the generalized $(L_0, L_1)$-smoothness assumption with heavy-tailed noises. They appreciate the new theoretical results. However, the paper requires major revision considering the following raised weaknesses: i) the discussion should be improved; the reviewers suggested the authors add a comparison of their results to those of previous methods under standard and light-tail noise assumptions; ii) the assumption on the noises should also be clarified; iii) the pessimistic dependence on $d$ raises concerns about the importance of this result; iv) the term "parameter-free" is not very accurate since the parameter-free method still depends on some parameters. The authors didn't engage in the rebuttal process, leaving the raised questions open. The paper is recommended for rejection.

**Reviewer Concerns:**

N/A: The authors didn't respond to the reviews

**Reviewer Scores:**

N/A: The authors didn't respond to the reviews

---

### Decision · Program_Chairs · 2026-01-26

Reject